# From Images to Signals: Are Large Vision Models Useful for Time Series Analysis?

## Abstract

Large Vision Models (LVMs) are emerging tools for transferring cross-modal knowledge to time series, but their potential for this domain is not yet fully understood. This work addresses the gap by investigating LVMs for both high-level (classification) and low-level (forecasting) time series tasks. Our aim is not only to assess whether LVMs can succeed, but also to reveal why they succeed or fall short. Through a comparative benchmark covering four representative LVMs, eight imaging methods, 18 datasets, and 21 baselines, we identify the strengths and limitations of the foundational LVMs, as well as effective strategies for adapting them to time series modeling. Our findings indicate that while the LVMs are effective for time series classification, they face notable challenges in forecasting. In particular, the best-performing LVM-based forecaster is limited to specific model types and imaging methods, exhibits biases toward periodic components in time series, and struggles to leverage long look-back windows. We hope our findings will serve as both a cornerstone and a practical guide for advancing future research on LVM- and multimodal-based solutions for different time series tasks.

## 1 Introduction

Time series analysis underpins applications in geoscience (Ardid et al., 2025), neuroscience (Caro et al., 2024), energy (Koprinska et al., 2018), healthcare (Morid et al., 2023), and smart cities (Ma et al., 2017). Inspired by advances in sequence modeling for language, recent research has explored methods ranging from Transformers (Wen et al., 2023) to Large Language Models (LLMs) (Jiang et al., 2024; Zhang et al., 2024) for time series analysis. With the success of Large Vision Models (LVMs) such as `ViT` (Dosovitskiy et al., 2021), `BEiT` (Bao et al., 2022), and `MAE` (He et al., 2022), emerging work has begun to investigate their potential in this domain (Chen et al., 2025). In these approaches, time series are *imaged*, *i.e.*, transformed into certain image representations (*e.g.*, Fig. 1(a)) (Ni et al., 2025), and then fed into an LVM to learn embeddings that can be probed for downstream tasks. The motivation for adapting LVMs, which are pre-trained on vast image corpora, to time series analysis rests on two perspectives: (1) for *high-level* (*i.e.*, pattern-level) tasks such as classification, imaged time series can encode distinguishable temporal patterns as semantic cues recognizable by LVMs; and (2) for *low-level* (*i.e.*, numerical-level) tasks such as forecasting, the structural similarity between images and time series – where rows or columns of *continuous* pixels in an image resemble a univariate time series (UTS) – makes LVMs more suitable than LLMs, which operate on *discrete* tokens. Despite this promise, the deeper connections between LVMs and time series analysis remain largely underexplored.

To understand the role of LVMs in time series tasks and inform future research – including multimodal models that integrate imaged time series (Zhong et al., 2025) – a thorough benchmarking study is needed. To this end, we investigate LVMs on two representative tasks: time series classification (TSC) and time series forecasting (TSF). In a nutshell, our conclusion is that **pre-trained LVMs prove versatile for TSC – a task that relies on pattern comparison – but remain constrained for TSF, which requires precise numerical inference**. Current best-performing LVM-based forecasters, although effective, remain confined to specific types of LVMs and imaging methods, exhibit biases toward forecasting periodic components in time series, and struggle with long look-back windows. We envision that our findings may also benefit other high-level tasks (*e.g.*, retrieval and clustering) as well as low-level tasks (*e.g.*, imputation and

anomaly detection). Unlike prior works that question the adoption of Transformers (Zeng et al., 2023a) and LLMs (Tan et al., 2024) in this field, we take a cautiously optimistic view, aiming to provide novel insights and practical caveats for selecting and adapting LVMs to appropriate time series tasks in order to support future developments.

This work involves two supervisedly pre-trained LVMs, *i.e.*, `ViT` (Dosovitskiy et al., 2021) and `Swin` (Liu et al., 2021), and two self-supervisedly pre-trained LVMs, *i.e.*, `MAE` (He et al., 2022) and `SimMIM` (Xie et al., 2022), along with eight widely used methods for imaging time series (Ni et al., 2025). The selected four LVMs cover key properties such as different pre-training strategies and attention mechanisms (detailed in §4.2), and underlie newer LVMs such as `ViT-22B` (Dehghani et al., 2023), `DINOv2` (Oquab et al., 2024), and `VIS-MAE` (Liu et al., 2024b). Another LVM, `LaVin-DiT` (Wang et al., 2025b), is also evaluated, but since it shows performance similar to that of the selected LVMs, its results are deferred to Appendix B.12. Our analysis involves ten datasets for TSC and eight datasets for TSF, all of which are widely used benchmarks (Bagnall et al., 2018; Wu et al., 2023; Tan et al., 2024). The results provide an overview of the effectiveness of LVMs, shedding light on which types of LVMs (*supervised vs. self-supervised*), imaging methods (*among the eight considered*), and output designs (*linear probing vs. pre-trained decoder*) are better suited for different tasks (*classification vs. forecasting*).

To uncover the true potential of LVMs for time series analysis, we conduct in-depth ablation studies. We compare their non-fine-tuned and fully/partially fine-tuned performance against that of the same architectures trained from scratch, thereby identifying the best adaptation strategies for TSC and TSF tasks, respectively. Experiments with shuffled time steps show that LVMs indeed capture temporal dependencies. As we observe that TSF is more challenging than TSC for LVMs, we further conduct a TSF-specific study, which reveals that the best-performing LVM-based forecaster is confined to a combination of a self-supervised LVM and a specific imaging method (*i.e.*, UVH in Fig. 1(a)). Intriguingly, we find that pre-trained decoders contribute more than encoders in forecasting, which helps explain the limitations of supervised LVMs. However, the current best-performing LVM-based forecasters exhibit an inductive bias: they tend to "combine past periods" to generate forecasts, causing them to favor datasets with strong periodicity. These observations highlight important directions for future improvement (§5). To summarize, our contributions are as follows:

- To the best of our knowledge, this is the first benchmark that comprehensively compares representative LVMs with time series models on both high-level and low-level tasks (§4.2).

- We summarize current best practices for adapting LVMs to time series analysis (§3) and conduct a series of ablation studies to assess whether LVMs are truly effective for TSC and TSF tasks. Our analysis covers various aspects of adapted LVMs, including pre-training, imaging, fine-tuning, architecture, decoding, temporal modeling, and computational cost (§4.3).

- We investigate the challenges of using LVMs for forecasting by examining individual model components, potential inductive biases, and the impact of look-back windows (§4.4).

## 2 Related Work

Our work shares similar merits with (Zeng et al., 2023a; Tan et al., 2024; Zhou & Yu, 2025), each of which sheds important light on a specific time series task, *i.e.*, Transformers for TSF (Zeng et al., 2023a), LLMs for TSF (Tan et al., 2024), and LLMs for time series anomaly detection (TSAD) (Zhou & Yu, 2025). In contrast, our work is specifically focused on LVMs, covers a broader range of tasks, and provides more in-depth analysis. This work can be regarded as a substantial complement to prior studies by adding a new perspective to our understanding of the role of large foundation models in the contemporary time series domain.

Vision models have been used for a variety of time series tasks, including classification (Li et al., 2023; Wu et al., 2023), forecasting (Zeng et al., 2023b; Yang et al., 2024), anomaly detection (Zhang et al., 2019; Wu et al., 2023), and generation (Li et al., 2022; Karami et al., 2024). Our work focuses on recent developments in using **pre-trained LVMs**, particularly Transformer-based models, for time series analysis. Image-pretrained CNNs have also been investigated in prior work (Namura et al., 2024; Li et al., 2020), but they are beyond the scope of this study due to their relatively smaller model sizes. To apply LVMs to time series, existing works typically transform time series into images using imaging methods (Ni et al., 2025). For example, `AST`

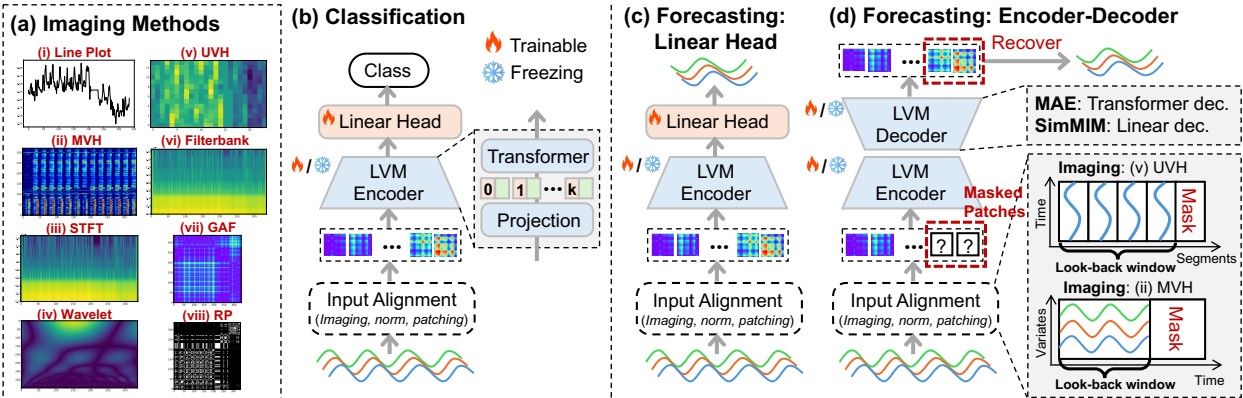

Figure 1: An overview of (a) different imaging methods, (b) LVM-based time series classification, (c) LVM with linear head for forecasting, (d) LVM encoder-decoder for forecasting. In (a), MVH encodes MTS, others encode UTS. (b)(c) apply to all LVMs (`ViT`, `Swin`, `MAE`, `SimMIM`) in this study. (d) applies to `MAE` and `SimMIM` with UVH/MVH images. Table 1 summarizes their applicability.

(Gong et al., 2021) applies ImageNet-pretrained `DeiT` (Touvron et al., 2021) to filterbank spectrograms of audio signals, which can essentially be regarded as UTS, for TSC. `ViTST` (Li et al., 2023) uses pre-trained `Swin` (Liu et al., 2021) to classify line plots of time series. These works have inspired a series of efforts to pre-train `ViT`-based architectures on imaged time series data, such as `SSAST` on AudioSet-2M (Gong et al., 2022), `ViTime` on synthetic data (Yang et al., 2024), and `Brain-JEPA` on brain time series (Dong et al., 2024). In contrast to TSC, the TSF task has received less attention in the context of LVMs, possibly because LVMs are less adept at low-level (numerical inference) tasks than at high-level (pattern recognition) tasks. The most notable method is `VisionTS` (Chen et al., 2025; Shen et al., 2025b), which adapts a self-supervisedly pre-trained LVM, *i.e.*, `MAE` (He et al., 2022), for zero-shot and few-shot TSF.

Recently, vision-language models (VLMs), such as `LLaVA` (Liu et al., 2023), `CLIP` (Radford et al., 2021), and `ViLT` (Kim et al., 2021), which involve pre-trained vision encoders, have been explored for TSC (Wimmer & Rekabsaz, 2023; Prithyani et al., 2024), TSAD (Zhuang et al., 2024), and TSF (Zhong et al., 2025). However, the effectiveness of standalone LVMs in time series analysis has not yet been well understood. Therefore, we focus on LVMs in this work, laying the groundwork for future investigations of VLMs in time series analysis (further discussion is deferred to in Appendix E).

## 3 Methods for Using LVMs in Time Series Analysis

This work does not aim to introduce new models. Instead, we assess **LVMs' innate ability** for time series analysis by keeping the main architecture intact while making a few **necessary modifications** for **cross-modal adaptation**, including (i) Input Alignment, and (ii) Task-Specific Augmentation. Additionally, we introduce two ablation studies that will be used in §4 to evaluate whether LVM architectures are overly complex. Fig. 1 provides an overview of the models used in this study.

**(i) Input Alignment.** The input to a pre-trained LVM is typically a normalized 3-channel image of a pre-defined size. Adapting time series data to the input format of LVMs requires: (1) converting the time series into images; (2) resizing the resulting images to satisfy the channel and size requirements; and (3) normalizing the images.

For (1), we employ eight imaging methods outlined by (Ni et al., 2025). As Fig. 1(a) illustrates, **Line Plot** draws a 2D image with the $x$-axis representing time steps and the $y$-axis representing time-wise values. **UVH** (Univariate Heatmap) divides a UTS, $\mathbf{x} \in \mathbb{R}^T$, into $\lfloor T/L \rfloor$ segments of length $L$ – where $L$ is a period obtained using Fast Fourier Transform (FFT) on $\mathbf{x}$ – and stacks them into a 2D image of size $L \times \lfloor T/L \rfloor$. **MVH** (Multivariate Heatmap) visualizes the matrix of a multivariate time series (MTS), $\mathbf{X} \in \mathbb{R}^{d \times T}$, with the $x$-axis representing the $T$ time steps and the $y$-axis representing the $d$ variates. **STFT** (Short-Time

Fourier Transform), **Wavelet** (Wavelet Transform), and **Filterbank** are three methods for transforming $\mathbf{x}$ into a **Spectrogram**, with the $x$-axis representing time and the $y$-axis representing frequency or scale. **GAF** (Gramian Angular Field) and **RP** (Recurrence Plot) produce square matrices with both $x$- and $y$-axis representing time, although they encode different temporal patterns. For brevity, we refer readers to (Ni et al., 2025) and our summaries in Appendix F.1 and Appendix F.2 for a more detailed introduction about the eight imaging methods.

For (2), *i.e.*, image resizing, following (Gong et al., 2021; Chen et al., 2025), we first resize an imaged time series to match the input size used in LVM pre-training via bilinear interpolation. We then adapt the resized images to satisfy the 3-channel requirement by duplicating each resized image (per variate) three times to form a grayscale image. For (3), *i.e.*, image normalization, since the adopted LVMs – namely, `ViT`, `Swin`, `MAE`, and `SimMIM` – standardize each pre-training image, we normalize each imaged time series in the same manner for consistency: $\mathbf{I}_{norm} = [\mathbf{I} - \text{mean}(\mathbf{I})]/\text{standard-deviation}(\mathbf{I})$, where $\mathbf{I}$ denotes the input image and $\mathbf{I}_{norm}$ denotes the normalized image. As shown in Fig. 1(b)-(d), the normalized image is then divided into a number of patches specified by each LVM before being fed into the model.

**(ii) Task-Specific Augmentation.** For the **TSC task**, as shown in Fig. 1(b), we probe each LVM's encoder. For `ViT` and `Swin`, this involves replacing their classification layers with a linear layer tailored to the downstream TSC task. For `MAE` and `SimMIM`, this involves replacing their reconstruction decoders with a linear classification layer. Since most imaging methods encode UTS (except MVH), the image of each variate is fed into the LVM individually. The output patch embeddings from all variates are concatenated before being passed to the final linear layer. For MVH, there is a single image covering all variates; thus, variate concatenation is not needed.

For the **TSF task**, we employ two frameworks from the literature. Fig. 1(c) trains a linear forecaster (Zeng et al., 2023b; Yang et al., 2024), while Fig. 1(d) uses LVM reconstruction decoders for forecasting (Chen et al., 2025). Because only `MAE` and `SimMIM` in our study have such decoders, Fig. 1(d) is applied to them, whereas Fig. 1(c) applies to `ViT` and `Swin`. For both frameworks, we adopt the widely used

Table 1: Summary of LVM-based frameworks.

| Task | Imaging | ViT | Swin | MAE | SimMIM |
|------|---------|-----|------|-----|--------|
| TSC | All | Fig. 1(b) | Fig. 1(b) | Fig. 1(b) | Fig. 1(b) |
| TSF | UVH, MVH | Fig. 1(c) | Fig. 1(c) | Fig. 1(d) | Fig. 1(d) |
| TSF | Other | Fig. 1(c) | Fig. 1(c) | Fig. 1(c) | Fig. 1(c) |

"variate-independence" assumption in TSF (Nie et al., 2023), *i.e.*, each variate is forecast independently. This assumption applies to all imaging methods except MVH, where all variates are visualized into a single image and thus forecast jointly. Additionally, the framework in Fig. 1(d) applies a mask after the look-back window portion of the image, then reconstructs the masked patches and recovers forecasts from the reconstructed patches. This requires input images to preserve raw (or normalized) time series values at the pixel level. Among the eight imaging methods, only MVH and UVH preserve time series values. Thus, this framework is applied to MVH and UVH[1]. The framework in Fig. 1(c) can be applied to all imaging types. Table 1 summarizes how frameworks (b), (c), and (d) in Fig. 1 are applied to different LVMs.

**Ablations.** To assess whether the LVM architecture is over-complex, we introduce two ablation models. Both models retain the projection layer in the LVM encoder but replace the Transformer with simpler layers. The first ablation replaces the Transformer with a linear layer, referred to as w/o-LVM. The second ablation uses a single randomly initialized multi-head attention layer, referred to as LVM2ATTN. Both ablations use a linear head to avoid complex decoders. They are applicable to all eight imaging types and both tasks. An illustration of these models is provided in Appendix B.5.

## 4 Experiments

**Datasets.** Our experiments are conducted on widely used benchmarks. For TSC, following (Wu et al., 2023; Zhou et al., 2023), we use ten datasets from the UEA Archive (Bagnall et al., 2018), covering

---

[1]GAF can also be applied because it has an inverse transform to recover time series (Wang & Oates, 2015), but its performance is largely limited for the factors described in Appendix B.2.

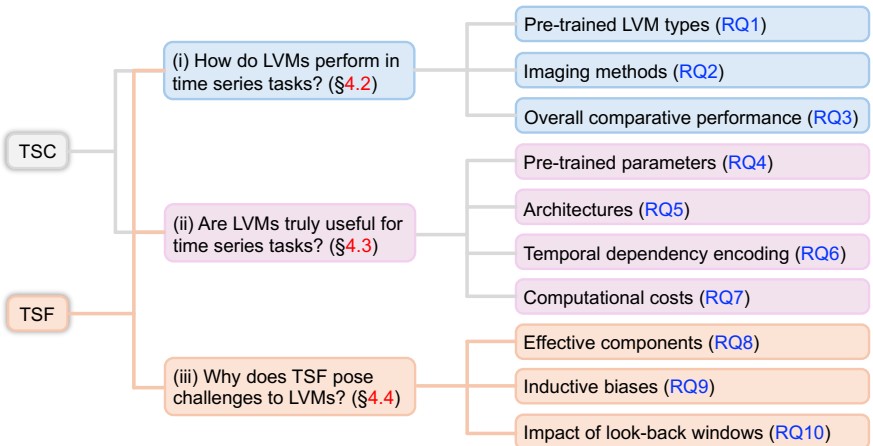

Figure 2: An overview of the structure of the experiments. "RQ" stands for research question.

gesture/action/audio recognition, heartbeat-based diagnosis, and other real-world tasks. For TSF, we use eight datasets, including the ETT (Electricity Transformer Temperature) benchmark (Zhou et al., 2021), which comprises ETTh1, ETTh2, ETTm1, and ETTm2, as well as the Weather (Wu et al., 2021), Illness (Wu et al., 2021), Traffic (Wu et al., 2021), and Electricity datasets (Trindade, 2015). For both tasks, all time series are MTS. More details about the datasets are provided in Appendix A.1.

**Evaluation Metrics.** For the TSC task, following (Wu et al., 2023; Zhou et al., 2023), we report classification accuracy for all compared methods. For the TSF task, following (Nie et al., 2023; Zeng et al., 2023a; Tan et al., 2024), mean squared error (MSE) and mean absolute error (MAE) are used to evaluate performance. The definitions of these evaluation metrics are provided in Appendix A.3.

**Compared Methods.** As discussed in §1, we base our experiments on representative LVMs, including two supervisedly pre-trained models: (1) `ViT` (Dosovitskiy et al., 2021) and (2) `Swin` (Liu et al., 2021), as well as two self-supervisedly pre-trained models: (3) `MAE` (He et al., 2022) and (4) `SimMIM` (Xie et al., 2022). These models are implemented in the frameworks as specified in Table 1 for different tasks. Following (Wu et al., 2023; Zhou et al., 2023), we include 18 classification baselines ranging from XGBoost to LLM-based methods. Moreover, we compare against eight SOTA forecasting baselines. Since our goal is to assess the **cross-modal knowledge** of LVMs, the baselines include models without pre-trained knowledge and models with other forms of cross-modal knowledge (*e.g.*, LLM-based models). The baselines are presented in Fig. 5 and Table 2, and are described in Appendix A.2. The implementation details, including checkpoint selection, hyperparameters, and runtime environments, are provided in Appendix A.5.

## 4.1 Structure of Experiments

Through a series of research questions (RQs), we assess (i) whether LVMs are effective for TSC and TSF tasks after selecting suitable configurations of *pre-trained LVM types* and *imaging methods* (RQ1-RQ3, §4.2); (ii) whether the best-performing LVMs (as identified in (i)) are truly useful in terms of details regarding their *pre-trained parameters*, *architectures*, *ability to encode temporal dependencies*, and *computational costs* (RQ4-RQ7, §4.3); and (iii) why TSF poses greater challenges for LVMs by investigating *effective model components*, *inductive biases*, and the *impact of look-back windows* (RQ8-RQ10, §4.4). Throughout (i) and (ii), we compare results between TSC and TSF to derive comparative insights that could generalize to pattern-level and numerical-level tasks. Fig. 2 summarizes the structure of the experiments.

It is noteworthy that (i) is not merely a reflection of existing work, as no prior study has formally compared *fine-tuned* LVMs across these two representative tasks. As additional contributions, the RQs also provide guidance on configuring LVMs – including *pre-trained LVM type selection* (RQ1), *imaging method selection* (RQ2), and *fine-tuning strategy selection* (RQ4) – for different time series tasks, and highlight future directions for improving LVM-based forecasters through the findings in (iii), as discussed in §5.

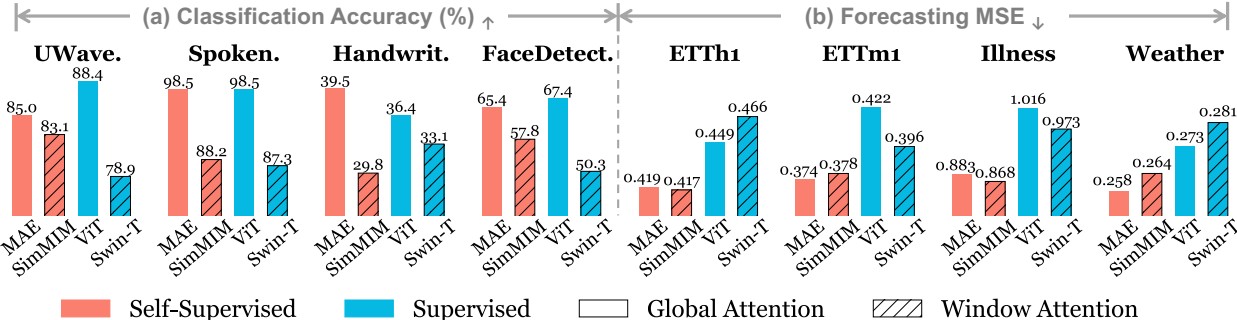

Figure 3: Comparison of four LVMs on the TSC (accuracy↑: higher is better) and TSF (MSE↓: lower is better) tasks. Two taxonomies of the LVMs: (1) supervised (`ViT`, `Swin`) *vs.* self-supervised (`MAE`, `SimMIM`), and (2) global attention (`ViT`, `MAE`) *vs.* window-based attention (`Swin`, `SimMIM`).

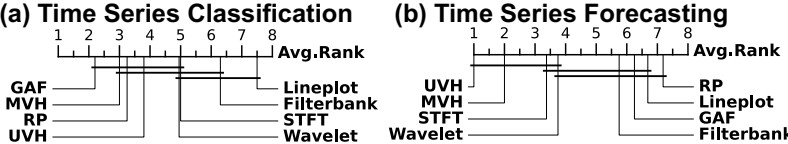

Figure 4: Average ranks of different imaging methods in (a) TSC task, and (b) TSF task. Lower ranks indicate better performance. The detailed results can be found in Appendix B.2.

## 4.2 Comparing LVMs with Non-LVM Methods: How Do LVMs Perform in Time Series Tasks?

This section first identifies the best combination of LVMs and imaging methods for the TSC and TSF tasks (RQ1 and RQ2), then compares the best settings with SOTA non-LVM baselines to assess the overall effectiveness of LVMs in time series tasks (RQ3).

**[RQ1] What type of LVM best fits the TSC (TSF) task?** Fig. 3 compares the four LVMs on TSC and TSF tasks. For conciseness, results from four UEA classification datasets (UWaveGestureLibrary, SpokenArabicDigits, Handwriting, and FaceDetection) and four forecasting datasets (ETTh1, ETTm1, Illness, and Weather) are presented.

From Fig. 3, we observe that (1) **supervised LVMs and self-supervised LVMs achieve comparable classification accuracies**, while (2) **self-supervised LVMs substantially outperform supervised LVMs in forecasting**. Observation (1) is consistent with the comparable performance of the two types of LVMs in image classification (He et al., 2022). Observation (2) can be attributed to the continuous nature of both pixels and time series, which enables self-supervised LVMs to transfer their capability of reconstructing masked pixels to predicting masked time series.

Moreover, in Fig. 3(a), we observe that `SimMIM` and `Swin` underperform (`SimMIM` uses a `Swin` backbone). This is likely because they employ a *window-based local attention* mechanism. Compared to the *global attention* adopted by `MAE` and `ViT`, local attention implicitly assumes *translation invariance* – the ability of a model to recognize an object regardless of its location in an image (Lenc & Vedaldi, 2015). However, this assumption does not hold for imaged time series, where different locations correspond to different time steps or frequencies. A pattern appearing at different time steps may correspond to different classes. By overlooking such spatial differences, `SimMIM` and `Swin` may fail to capture some time- or frequency-sensitive patterns.

**[RQ2] Which imaging method best fits the TSC (TSF) task?** Fig. 4 presents the critical difference (CD) diagrams (Han et al., 2022) of the average ranks of the eight imaging methods on TSC and TSF tasks (lower ranks indicate better performance). From Fig. 4(a), **GAF performs best for classification**, followed by MVH and RP, indicating their ability to encode distinguishable temporal patterns. In contrast, Line Plot substantially underperforms and therefore may be less suitable for this task. **For forecasting**, Fig. 4(b) shows that, when combined with the reconstruction framework in Fig. 1(d), **UVH achieves the**

Table 2: Model comparison on TSF task. The results are averaged over different forecasting horizons. See Table 15 in Appendix B.3 for full results. **Red** and blue numbers are the the best and second best results in each column. # Wins is the number of times each method performs best.

| Method | MAE | | ViT | | Time-LLM | | GPT4TS | | CALF | | Dlinear | | PatchTST | | TimesNet | | FEDformer | | Autoformer | |
|---|---|---|---|---|---|---|---|---|---|---|---|---|---|---|---|---|---|---|---|---|---|
| Metrics | MSE | MAE | MSE | MAE | MSE | MAE | MSE | MAE | MSE | MAE | MSE | MAE | MSE | MAE | MSE | MAE | MSE | MAE | MSE | MAE |
| ETTh1 | **0.409** | **0.419** | 0.445 | 0.449 | 0.418 | 0.432 | 0.418 | 0.421 | 0.432 | 0.431 | 0.423 | 0.437 | 0.413 | 0.431 | 0.458 | 0.450 | 0.440 | 0.460 | 0.496 | 0.487 |
| ETTh2 | 0.357 | 0.390 | 0.389 | 0.411 | 0.361 | 0.396 | 0.354 | 0.389 | 0.351 | 0.384 | 0.431 | 0.447 | **0.330** | **0.379** | 0.414 | 0.427 | 0.437 | 0.449 | 0.450 | 0.459 |
| ETTm1 | **0.345** | **0.374** | 0.409 | 0.422 | 0.356 | 0.377 | 0.363 | 0.378 | 0.396 | 0.391 | 0.357 | 0.379 | 0.351 | 0.381 | 0.400 | 0.406 | 0.448 | 0.452 | 0.588 | 0.517 |
| ETTm2 | 0.268 | 0.327 | 0.300 | 0.337 | 0.261 | 0.316 | **0.254** | **0.311** | 0.283 | 0.323 | 0.267 | 0.334 | 0.255 | 0.315 | 0.291 | 0.333 | 0.305 | 0.349 | 0.327 | 0.371 |
| Weather | **0.225** | 0.258 | 0.234 | 0.273 | 0.244 | 0.270 | 0.227 | **0.255** | 0.251 | 0.274 | 0.249 | 0.300 | 0.226 | 0.264 | 0.259 | 0.287 | 0.309 | 0.360 | 0.338 | 0.382 |
| Illness | 1.837 | 0.883 | 2.179 | 1.016 | 2.018 | 0.894 | 1.871 | 0.852 | 1.700 | 0.869 | 2.169 | 1.041 | **1.443** | **0.798** | 2.139 | 0.931 | 2.847 | 1.144 | 3.006 | 1.161 |
| Traffic | **0.386** | **0.256** | 0.430 | 0.343 | 0.422 | 0.281 | 0.421 | 0.274 | 0.444 | 0.284 | 0.434 | 0.295 | 0.391 | 0.264 | 0.620 | 0.336 | 0.610 | 0.376 | 0.628 | 0.379 |
| Electricity | **0.159** | **0.250** | 0.173 | 0.266 | 0.165 | 0.259 | 0.170 | 0.263 | 0.176 | 0.266 | 0.166 | 0.264 | 0.162 | 0.253 | 0.193 | 0.295 | 0.214 | 0.327 | 0.227 | 0.338 |
| # Wins | 9 | | 0 | | 0 | | 3 | | 0 | | 0 | | 4 | | 0 | | 0 | | 0 | |

**best performance**, followed by MVH. This suggests their suitability for numerical-level tasks by leveraging the knowledge acquired by LVMs during masked-pixel reconstruction in large-scale image-based pre-training (further analysis is provided in Appendix B.2).

**[RQ3] Can LVMs achieve SOTA performance?** Fig. 5 and Table 2 present the overall performance of the compared methods, where `ViT` and `MAE` are selected as the best-performing supervised and self-supervised LVMs, respectively. Here, `ViT` and `MAE` are configured with their best-performing imaging methods – GAF for TSC and UVH for TSF. On average, the LVMs are fine-tuned on each dataset for 20 epochs for TSC and eight epochs for TSF with early stopping. Our experiments follow standard evaluation protocols of TSC (Zhou et al., 2023) and TSF (Tan et al., 2024).

From Fig. 5, both `ViT` and `MAE` outperform the baselines, suggesting **both supervised and self-supervised LVMs are effective for high-level (*i.e.*, pattern level) TSC task**. This is consistent with their ability to classify regular images (He et al., 2022). From Table 2, across the eight datasets and two metrics, `MAE` outperforms non-LVM baselines in 9/16 cases, while `ViT` does not show clear superiority over non-LVM baselines, which may be due to its classification-based pre-training. The results suggest **distinct behaviors of LVMs in TSF, indicating that greater challenges may arise in low-level (*i.e.*, numerical level) tasks**. A closer look at Table 2 shows that, despite `ViT`'s inferior performance, it remains comparable to `DLinear` in many cases.

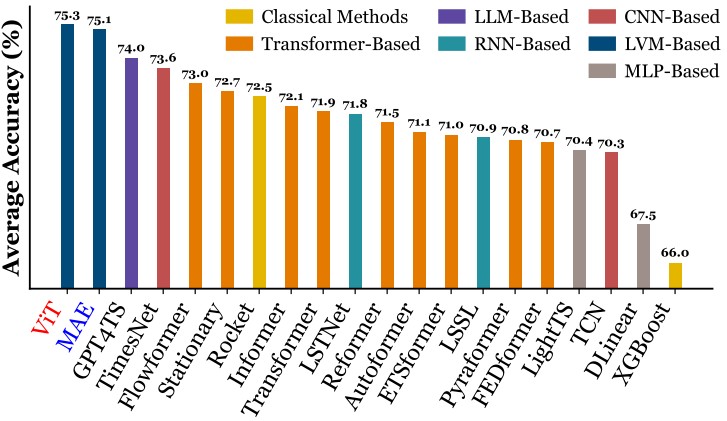

Figure 5: Model comparison on TSC task. The results are averaged over ten UEA datasets. The full results are in Appendix B.3.

This suggests that although `ViT` is pre-trained for image classification, linearly probing is sufficient to produce reasonable forecasting results, indicating potential for cross-task/modality knowledge transfer.

---

**Key Insights from RQ1 - RQ3 (§4.2)**

- Both supervised and self-supervised LVMs are effective for TSC, whereas self-supervised LVMs are better suited to TSF than supervised LVMs, implying TSF may pose greater challenges for LVMs.
- The imaging method GAF performs best for TSC, whereas for TSF, UVH performs best.
- The best LVM configurations – `ViT` with GAF imaging for TSC and `MAE` with UVH imaging for TSF – achieve SOTA performance compared with non-LVM baselines.

Table 3: Ablation studies. For the TSC task, higher accuracy indicates better performance. For the TSF task, lower MSE indicates better performance. The full results are provided in Appendices B.4 and B.5.

| Task | | TSC Task (accuracy (%)$_\uparrow$) | | | | TSF Task (MSE$_\downarrow$) | | | |
|---|---|---|---|---|---|---|---|---|---|
| **Dataset** | | UWave. | Spoken. | Handwrit. | FaceDetect. | ETTh1 | ETTm1 | Illiness | Weather |
| RQ4 | (a) Fine-tuning all parameters | 88.4 | **98.5** | **36.4** | **67.4** | 0.558 | 0.399 | 1.781 | 0.273 |
| | (b) Fine-tuning all but `CLS` & `Mask` | 87.5 | 98.2 | 35.2 | 66.3 | 0.530 | 0.408 | 1.783 | 0.275 |
| | (c) Fine-tuning MLP & norm | **88.7** | 98.4 | 35.5 | 67.1 | 0.532 | 0.396 | 1.737 | 0.264 |
| | (d) Fine-tuning norm | 81.6 | 98.0 | 28.5 | 65.2 | **0.409** | **0.345** | 1.837 | **0.225** |
| | (e) Freezing all parameters | 84.0 | **98.5** | 27.8 | 66.7 | 0.452 | 0.420 | 2.037 | 0.308 |
| | (f) Training from scratch | 73.4 | 97.0 | 24.3 | 65.0 | 0.475 | 0.372 | **1.723** | 0.241 |
| RQ5 | w/o-LVM | 78.6 | 96.4 | 22.4 | 64.1 | 0.423 | 0.376 | 2.291 | 0.255 |
| | LVM2Attn | 80.1 | 96.5 | 20.7 | 66.2 | 0.428 | 0.357 | 2.108 | 0.254 |

### 4.3 Are LVMs Truely Useful for Time Series Tasks?

This section aims to uncover whether the effectiveness of LVMs observed in Fig. 5 and Table 2 truly stems from their pre-trained parameters, architectures, and temporal modeling capabilities. Additionally, we assess the computational costs of LVMs as part of their practical applicability. The following analyses use four UEA classification datasets (UWaveGestureLibrary, SpokenArabicDigits, Handwriting, and FaceDetection) and four forecasting datasets (ETTh1, ETTm1, Illness, and Weather) for conciseness. Unless otherwise noted, the best-performing LVM is used for each task: `ViT` with GAF imaging for TSC (*cf.* Fig. 5) and `MAE` with UVH imaging for TSF (*cf.* Table 2).

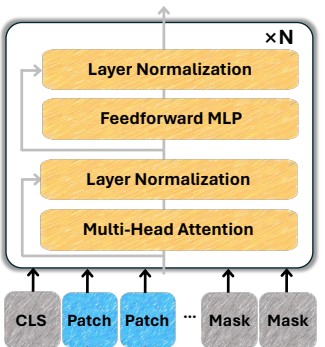

Figure 6: Key Components in LVM Transformer block.

**[RQ4] Are the pre-trained parameters in LVMs useful in time series tasks?** To answer this question, we compare three types of ablations: (1) training LVMs from scratch; (2) freezing all LVM parameters; and (3) fine-tuning LVMs for a few epochs. Since different tasks may require different fine-tuning strategies, we further conduct a series of fine-tuning ablations that progressively freeze the key components within the Transformer block of LVMs. Fig. 6 illustrates these components. Specifically, our ablations include: (a) fine-tuning all parameters; (b) fine-tuning all parameters except the `CLS` and `Mask` tokens; (c) fine-tuning only the MLP and normalization layers; (d) fine-tuning only the normalization layers; (e) freezing all parameters; and (f) randomly initializing an LVM and training it from scratch.

Table 3 (upper panel) summarizes the results. For TSC, we observe that "freezing all parameters" setting outperforms "training from scratch" in all cases, suggesting that **LVMs indeed transfer useful knowledge**. Fine-tuning all parameters for a few epochs consistently improves upon the "freezing all parameters" setting, further validating the effectiveness of knowledge transfer. Comparing (a) and (c), **we find that fine-tuning only the MLP and normalization layers is the minimal effort required to achieve performance comparable to full fine-tuning in TSC.**

For TSF, surprisingly, neither the "freezing all parameters" setting nor the "fine-tuning all parameters" setting consistently outperforms "training from scratch". In contrast, **fine-tuning only the normalization layers significantly improves performance in TSF**. This may be attributed to the low-level nature of forecasting tasks. Since the model must predict numerical values directly, performance is strongly influenced by normalization, whereas fine-tuning more parameters than necessary may lead to overfitting. This contrasts with classification tasks, where learning high-level temporal patterns involves more components that normalization layers alone, making the fine-tuning of additional parameters beneficial.

**[RQ5] How useful are LVM architectures?** In [RQ4], training LVMs from scratch may lead to overfitting on small training datasets due to their complex architectures. To investigate this, we evaluate the two simpler models introduced in §3, namely w/o-LVM and LVM2Attn. From Table 3 (bottom panel), We observe

Table 4: Performance drops of the compared models under different temporal perturbations. Red color marks the largest drop under each perturbation strategy. The full results are provided in Appendix B.6.

| Task | | TSC Task | | | | TSF Task | | | |
|---|---|---|---|---|---|---|---|---|---|
| Dataset | | UWave. | Spoken. | Handwrit. | FaceDetect. | ETTh1 | ETTm1 | Illiness | Weather |
| **Sf-All** | w/o-LVM | 78.2% | 49.7% | 81.7% | 19.3% | 76.2% | 98.4% | 116.4% | 24.1% |
| | LVM2Attn | 86.4% | 50.6% | 89.9% | 22.4% | 79.7% | 117.1% | 109.1% | 24.4% |
| | LVM | 80.7% | 84.7% | 91.5% | 29.2% | 83.8% | 118.4% | 162.8% | 44.5% |
| **Sf-Half** | w/o-LVM | 6.6% | 12.4% | 74.6% | 10.8% | 14.4% | 28.3% | 41.6% | 2.4% |
| | LVM2Attn | 8.7% | 11.6% | 83.6% | 11.3% | 19.5% | 44.8% | 69.3% | 2.4% |
| | LVM | 36.4% | 30.2% | 86.5% | 9.3% | 14.5% | 48.2% | 21.3% | 9.6% |
| **Ex-Half** | w/o-LVM | 98.8% | 82.2% | 83.5% | 22.8% | 13.0% | 145.3% | 11.0% | 34.0% |
| | LVM2Attn | 98.9% | 82.3% | 87.0% | 24.6% | 9.1% | 158.3% | 27.9% | 35.5% |
| | LVM | 59.4% | 89.9% | 97.0% | 9.2% | 14.2% | 242.3% | 23.0% | 67.2% |
| **Masking** | w/o-LVM | -1.0% | 3.1% | 22.3% | -1.2% | 47.3% | 58.5% | 94.1% | 33.4% |
| | LVM2Attn | 1.0% | 3.6% | 20.3% | 2.7% | 46.0% | 70.3% | 127.8% | 33.6% |
| | LVM | 29.0% | 41.8% | 56.0% | 7.4% | 47.5% | 58.4% | 128.9% | 49.6% |

Table 5: Computational costs of LVMs and the two best-performing baselines in TSC (`GPT4TS`, `TimesNet`) and TSF (`PatchTST`, `GPT4TS`) tasks. The TSF costs are measured with a forecasting horizon of 96 time steps.

| Method | | LVM | | | 1st Baseline (task specific) | | | 2nd Baseline (task specific) | | |
|---|---|---|---|---|---|---|---|---|---|---|
| Task | Dataset | # Param (M) | Train (min) | Inference(ms) | # Param (M) | Train (min) | Inference(ms) | # Param (M) | Train (min) | Inference(ms) |
| **TSC** | UWave. | 89.43 | 2.83 | 11.52 | 82.23 | 1.19 | 57.61 | 2.42 | 0.39 | 1.69 |
| | Handwrit. | 97.59 | 5.18 | 23.72 | 83.62 | 1.33 | 50.51 | 2.47 | 0.51 | 0.78 |
| **TSF** | ETTh1 | 111.91 | 9.99 | 4.32 | 3.75 | 0.52 | 0.18 | 85.02 | 10.46 | 0.50 |
| | Weather | 111.91 | 207.83 | 1.50 | 6.90 | 16.97 | 0.10 | 86.64 | 94.10 | 0.35 |

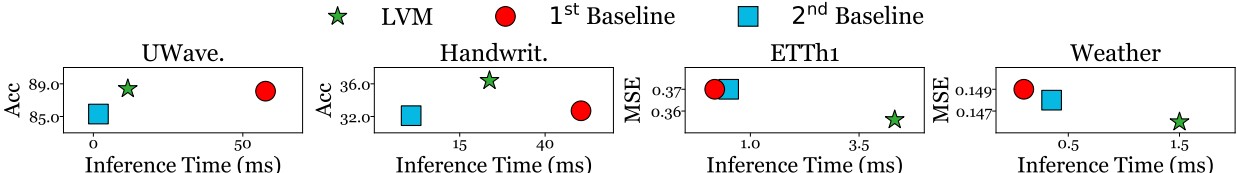

Figure 7: Inference time *vs.* performance of the compared methods in TSC task (accuracy↑) using UWaveGesture and Handwriting datasets, and in TSF task (MSE↓) using ETTh1 and Weather datasets. The full results are provided in Appendix B.7.

that training LVMs from scratch does not consistently outperform these simpler models, suggesting that the architecture of LVMs alone may be overly complex for the available training data. However, since training from scratch is generally no worse than the simpler models, the overfitting issue does not appear to be severe. Moreover, the "freezing all parameters" setting and all fine-tuning settings (a)-(d) outperform the simpler models in TSC, while fine-tuning setting (d) consistently outperforms the simpler models in TSF. These results indicate that **LVM architectures are not overly complex when serving as containers of transferrable knowledge acquired through pre-training**.

**[RQ6] Can LVMs capture temporal dependencies in time series?** Temporal order plays a critical role in time series analysis. It is of significant interest to understand whether LVMs can capture temporal dependencies. To this end, following (Tan et al., 2024), we perturb the temporal order using four methods: (1) **Sf-All**, which randomly shuffles all time points; (2) **Sf-Half**, which randomly shuffles the first half of the time points; (3) **Ex-Half**, which swaps the first and second halves of the time points; and (4) **Masking**, which randomly masks 50% of the time points. Table 4 summarizes the relative performance drops. Following (Zeng et al., 2023a; Tan et al., 2024), the ablation models are used as baselines. From Table 4, we observe that LVMs consistently exhibit performance drops under temporal perturbations. Moreover, they are more vulnerable to these perturbations than the ablation models. This suggests that **LVMs likely make effective use of temporal patterns in time series during inference**.

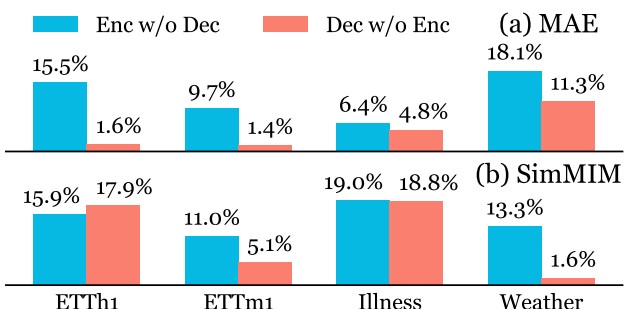

Figure 8: TSF performance drop (%) of (a) `MAE` and (b) `SimMIM` when initializing the encoder with pre-trained parameters (blue) and initializing the decoder with pre-trained parameters (red), respectively.

**[RQ7] What are the computational costs of LVMs?** We evaluate the training and inference time of LVMs. Training time is measured until model convergence under early stopping. Inference time is estimated by the average runtime per test sample. Table 5 compares LVMs with the best-performing baselines in TSC task (Fig. 5), namely `GPT4TS` and `TimesNet`, and in TSF task (Table 2), namely `PatchTST` and `GPT4TS`. From Table 5, we observe that LVMs have a larger number of parameters than the baselines. On average, LVMs require 3x (16x) training time of the best TSC (TSF) baseline, primarily due to the larger number of trainable parameters. For inference, LVMs are 4x faster than the best TSC baseline but 20x slower than the best TSF baseline. This is due to both the increased model size and the additional cost of transforming time series into images. Fig. 7 shows inference time *vs.* performance. Compared with the best baselines, **LVMs achieve reasonable computational costs in TSC but trade increased computational overhead for better performance in TSF**.

> **Key Insights from RQ4 - RQ7 (§4.3)**
>
> - The pre-trained parameters of LVMs encode transferrable knowledge that benefits TSC and TSF.
> - For TSC, fine-tuning MLP & norm layers is sufficient. For TSF, fine-tuning norm layers is sufficient.
> - LVM architectures are not overly complex when serving as containers of transferrable knowledge.
> - LVMs demonstrate the ability to capture temporal dependencies in time series during inference.
> - The computational costs of LVMs remain reasonable for TSC but are relatively higher for TSF.

### 4.4 Why Does Time Series Forecasting Pose Challenges to LVMs?

This section investigates the underlying causes of the challenges observed in LVM-based forecasting in §4.2.

**[RQ8] Which component of LVMs contributes more to TSF?** Usually, pre-trained encoders are regarded as general feature extractors and are widely used for knowledge transfer, whereas decoders are often task-specific and therefore re-trained for downstream tasks. However, the intuition appears to be reversed when adapting LVMs to TSF. Fig. 8 shows the performance degradation of two ablations relative to `MAE` and `SimMIM`: (1) **Enc w/o Dec**, which fine-tunes the pre-trained encoder while re-training a randomly initialized decoder; and (2) **Dec w/o Enc**, which fine-tunes the pre-trained decoder while re-training a randomly initialized encoder. Both ablations are trained until convergence. As shown in Fig. 8, **Enc w/o Dec** exhibits a larger performance drop than **Dec w/o Enc**, suggesting that **pre-trained decoders contribute more to TSF performance than pre-trained encoders**. This can be attributed to the fact that LVM decoders are pre-trained to reconstruct pixel values, making them naturally suited to the numerical-level TSF task (see Appendix E for further analysis). Notably, the decoder of `SimMIM` is merely a linear layer accounting for only 3.8% of the model parameters, yet its contribution surpasses that of the substantially larger encoder, further highlighting the importance of pre-trained decoders for TSF. This observation also helps explain **the difficulty of adapting supervised LVMs to TSF (Fig. 3(b)), as they typically lack pre-trained decoders for numerical inference**. Additionally, it suggests **a promising future direction: pre-training more effective LVM decoders to improve TSF performance.**

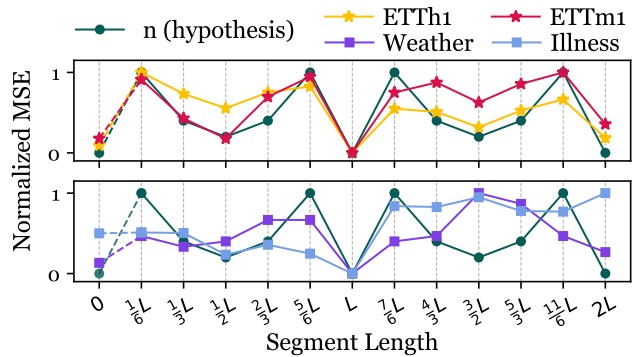

Figure 9: TSF performance of `MAE` *w.r.t.* varying segment length used in the UVH imaging method across datasets. "$n$" (green) estimates TSF difficulty, as described in RQ9.

**[RQ9] Does the period-based UVH imaging method introduce any bias in TSF?** In Table 2, the best-performing LVM forecaster is `MAE` with UVH imaging. As illustrated in Fig. 1(a)(d), UVH is a period-based imaging method that stacks length-$L$ segments of a UTS $\mathbf{x}$ into a 2D image of size $L \times \lfloor T/L \rfloor$, where $L$ denotes the period. **We find that this method induces an inductive bias toward "forecasting periods".** In Fig. 9, we evaluate `MAE`'s forecasting performance by varying the segment length from $\frac{1}{6}L$ to $\frac{12}{6}L$, with the MSE

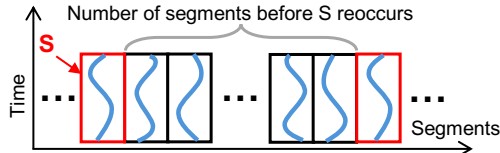

Figure 10: An illustration of UVH imaging.

values min-max normalized to the range $[0, 1]$. An estimated MSE value at segment length 0 is included by averaging the MSEs at $L$ and $2L$, since a segment length of zero is not computable. This point (together with the green lines) will be used later in the analysis. As shown in Fig. 9, `MAE` achieves its best performance at $L$ and $2L$, suggesting that (1) the datasets exhibit strong periodicity, and (2) `MAE` tends to forecast future values by "combining" past periodic segments. When the segment length does not align with the underlying periods, *i.e.*, when it differs from $L$ or $2L$, `MAE`'s forecasting accuracy deteriorates substantially.

Interestingly, under the UVH imaging framework, we can estimate the difficulty of TSF for `MAE` using the segment length. Specifically, the forecasting difficulty is highly correlated with *how frequently a segment reoccurs*, which can be quantified by the number of intervening segments between two consecutive occurrences, as illustrated in Fig. 10. When the two occurrences are farther apart, it becomes more difficult for `MAE` to capture the underlying periodic patterns. More formally, suppose the UTS is divided into segments of length $\frac{i}{k}L$. For example, in Fig. 9, $k = 6$, and $i \in [1, ..., 12]$. The following lemma characterizes the number of segments that elapse before a given segment reoccurs.

**Lemma 4.1.** *Let $\mathbf{x}$ be a UTS with an exact period $L$, i.e., $\mathbf{x}_t = \mathbf{x}_{t+L}$. Suppose $\mathbf{x}$ is divided into segments of length $\frac{i}{k}L$, where $i, k \in \mathbb{N}^+$. Then, the smallest number of segments, $n$, before a segment reoccurs, i.e., $\mathbf{x}_t = \mathbf{x}_{t+n \cdot (i/k)L}$, is given by $n = \frac{k}{GCD(i,k)}$, where $GCD(\cdot, \cdot)$ denotes the greatest common divisor.*

The proof of Lemma 4.1 is provided in Appendix C. Lemma 4.1 states that $n$ can be computed given $i$ and $k$. To validate the correlation between $n$ and TSF difficulty, we compute $n$ and plot it in Fig. 9, after normalizing it to the range $[0, 1]$. According to Lemma 4.1, $n$ is small when $\frac{i}{k} = 1$ or 2, yielding $n = 1$. When $\frac{i}{k} = \frac{1}{2}$ or $\frac{3}{2}$, $n = 2$. This results in an "M"-shaped curve (green) in Fig. 9. The strong alignment between this curve and the MSEs on the ETTh1 and ETTm1 datasets supports our estimation of TSF difficulty, suggesting that `MAE` with UVH imaging has an inductive bias toward periodicity. In contrast, the MSEs on the Weather and Illness datasets align less well with the $n$-values, likely due to their weaker periodicity. Further analysis on datasets of weak periodicity is provided in Appendix B.10.

**[RQ10] Can LVMs make effective use of look-back windows?** Intuitively, longer look-back windows can facilitate forecasting (Zeng et al., 2023a). We evaluate `MAE` (with UVH imaging) using different look-back window lengths in Fig. 11. The Illness dataset is excluded because its time series are relatively short (966 time steps in total across the entire dataset). As shown in Fig. 11, `MAE`'s **performance improves as**

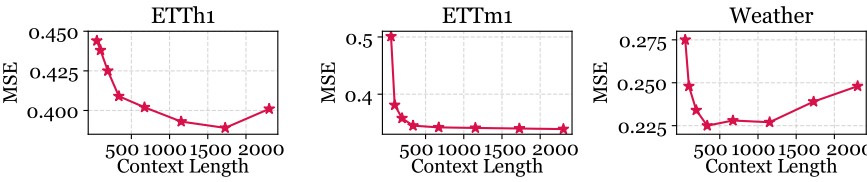

Figure 11: TSF performance (MSE) of `MAE` with varying look-back window (or context) lengths.

**the look-back window increases up to 1000 time steps, after which it plateaus or declines**. This behavior may stem from the image transformation process. Since pre-trained LVMs operate on fixed-size images with a limited number of pixels, they may be unable to capture all relevant information from excessively long time series. Moreover, image pixels are not fully utilized by the UVH imaging method to encode time-step information, as further analyzed in Appendix E. **This observation highlights a future research direction: improving pixel utilization for encoding time series information in TSF**. Fortunately, contemporary LVMs can effectively handle moderately long look-back windows, and a length of 1000 time steps is sufficient for many applications. Future models may further extend this capability.

---

**Key Insights from RQ8 - RQ10 (§4.4)**

- For optimally configured LVM-based forecasters, namely self-supervised LVMs with UVH imaging, pre-trained decoders contribute more to TSF performance than pre-trained encoders.
- SOTA LVM-based forecasters exhibit an inductive bias that favors datasets with strong periodicity.
- The UVH imaging method does not fully utilize image pixels. The limited number of pixels and their incomplete utilization reduce the ability of LVM-based forecasters to capture all relevant information in excessively long time series.

---

## 5 Conclusion and Future Directions

In this work, we explore the potential of LVMs for time series analysis in both high-level (TSC) and low-level (TSF) tasks. Through extensive experiments across four representative types of LVMs, eight imaging methods, 18 datasets, and 26 baselines, we provide insights into whether image-pretrained LVMs are effective for time series tasks, how to configure them for optimal performance, and what challenges remain in LVM-based TSF. These insights can facilitate the adoption of LVMs for time series analysis across research and applications.

Meanwhile, our benchmark study highlights several promising directions for future research: (1) integrating different imaging methods to combine their distinct encoding capabilities; (2) reducing inference latency in LVM-based TSF; (3) enhancing encoder utilization in LVM-based TSF; (4) pre-training more effective decoders for LVM-based forecasters; (5) addressing or harnessing the inductive bias of LVM-based forecasters; and (6) improving pixel utilization for encoding time series information in TSF. Moreover, we observe that a considerable portion of current TSF benchmark datasets exhibit strong periodicity, which may lead to biased conclusions about models with inductive biases similar to those of SOTA LVM-based forecasters. Therefore, diversifying benchmark datasets is also an important future research direction.

Finally, it is worth noting that time series foundation models (TSFMs) such as `Chronos` (Ansari et al., 2025), `Moirai` (Liu et al., 2025a), and `LightGTS` (Wang et al., 2025a) encode *intra-modal knowledge*, *i.e.*, knowledge acquired from large-scale time series pre-training, which is complementary to the *cross-modal (visual) knowledge* encoded by LVMs. For example, a promising future direction is to fine-tune image-pre-trained LVMs on large-scale time series data, thereby integrating both cross-modal and intra-modal knowledge to further enhance performance. Consequently, TSFMs are not considered direct competitors to LVMs in terms of the knowledge they encode and are therefore not included as major baselines. However, a comparison with `LightGTS` in Appendix G suggests a strong potential of the cross-modal knowledge encoded by LVMs.

We hope this study complements existing research and lays the groundwork for future work on multimodal, agentic time series analysis.

## 6 Discussion about Limitations

The use of LVMs for time series analysis is an emerging area of active research. This work serves as a pioneering effort in investigating the role of LVMs in both high-level and low-level time series tasks. As an initial exploration, we acknowledge some limitations in this work.

First, the key novelty of this research lies in uncovering the strengths and weaknesses of LVMs and providing in-depth analyses of their applicability to time series problems. As we are not proposing new models but tweaking the existing LVMs to fit time series tasks, there is limited novelty in model development. Second, there are more tasks in time series analysis than classification and long-term forecasting, such as short-term forecasting, imputation, retrieval, and anomaly detection. To ensure in-depth analyses, we only include the most representative tasks from the high-level and low-level task groups for clarity. However, most of our analyses are not task-specific, such as the evaluation of LVMs' potential in §4.2 and the analyses in RQ1-RQ7, which are likely generalizable to other tasks in the high-level group, including retrieval, and the low-level group, including short-term forecasting, imputation, and anomaly detection. It is noteworthy that tasks in the same group share similar objectives. For example, both classification and retrieval require recognition of similar/dissimilar time series. As another example, both forecasting and imputation require prediction of numerical values in time series. As such, our analyses of LVMs' ability in dealing with these objectives could be generally useful. Third, although 18 benchmark datasets have been used by following prior works involving TSC and TSF (Wu et al., 2023; Zhou et al., 2023; Zeng et al., 2023a), we are aware of other datasets such as those in UCR archive for TSC and GIFT-Eval for TSF (Aksu et al., 2024). Incorporating these additional datasets could further strengthen our analysis. However, we consider our findings from the 18 benchmark datasets solid and well grounded for their wide coverage of domains and patterns. Fourth, following prior works that evaluate multiple tasks (Wu et al., 2023; Zhou et al., 2023), we average of the results of TSC over different datasets in Fig. 5 and the results of TSF over different forecasting horizons in Table 2. The averaged results may be biased and may not be the best way to summarize the comparison of different methods. As such, we consider these results as an overview of LVMs' potential in TSC and TSF. We encourage inspection of the full results provided in Table 13 and Table 15 in Appendix B.3 for accurate assessment. Finally, as discussed in §2 and Appendix E, existing vision-language models (VLMs), such as `LLaVA`, `CLIP`, `ViLT`, and large multimodal models (LMMs), such as `Gemini`, `GPT-4o`, and `Claude-3`, have incorporated pre-trained vision encoders that are analogous to LVMs. Since the effectiveness of sole LVMs in time series analysis has not yet been well studied, in this work, we focus our research on LVMs, which can lay the groundwork for future research on VLMs and LMMs in time series analysis.

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

# A  Experimental Setup

## A.1  Benchmark Datasets

**Time Series Classification**. For TSC, following (Wu et al., 2023; Zhou et al., 2023), our experiments are conducted on 10 multivariate benchmark datasets from UEA archive (Bagnall et al., 2018), which span diverse domains, including chemical analysis, cognitive neuroscience, gesture recognition, biomedical signal processing, speech recognition and traffic analysis. The datasets are preprocessed following (Zerveas et al., 2021). Table 6 summarizes the statistics of the datasets.

Table 6: Statistics of the datasets for TSC. "Dataset Size" is organized in (Train, Validation, Test sets).

| Dataset | # Variates | Series Length | Dataset Size | Classes |
|---|---|---|---|---|
| EthanolConcentration | 3 | 1751 | (261, 263, 263) | 4 |
| FaceDetection | 144 | 62 | (5890, 3524, 3524) | 2 |
| Handwriting | 3 | 152 | (150, 850, 850) | 26 |
| Heartbeat | 61 | 405 | (204, 205,205) | 2 |
| Japanese Vowels | 12 | 29 | (270, 370, 370) | 9 |
| PEMS-SF | 963 | 144 | (267, 173, 173) | 7 |
| SelfRegulationSCP1 | 6 | 896 | (268, 293, 293) | 2 |
| SelfRegulationSCP2 | 7 | 1152 | (200, 180, 180) | 2 |
| SpokenArabicDigits | 13 | 93 | (6599, 2199, 2199) | 10 |
| UWaveGestureLibrary | 3 | 315 | (120, 320, 320) | 8 |

**Time Series Forecasting**. For TSF, following (Zhou et al., 2021; Wu et al., 2021; Nie et al., 2023; Zeng et al., 2023a; Tan et al., 2024; Chen et al., 2025), our experiments are conducted on 8 widely used benchmark datasets. The four ETT datasets (ETTh1, ETTh2, ETTm1, ETTm2) record oil temperature from two electric transformers, sampled at 15-minute and hourly intervals. The Weather dataset collects measurements of meteorological indicators in Germany every 10 minutes. The Illness dataset keeps weekly counts of patients and the influenza-like illness ratio from the United States. The Traffic dataset measures hourly road occupancy rates from sensors on San Francisco freeways. The Electricity dataset records hourly electricity consumption of Portuguese clients. Table 7 summarizes the statistics of the datasets.

Table 7: Statistics of the datasets for TSF. "Dataset Size" is organized in (Train, Validation, Test sets).

| Dataset | # Variates | Series Length | Dataset Size | Frequency |
|---|---|---|---|---|
| ETTh1 | 7 | 17420 | (8545, 2881, 2881) | Hourly |
| ETTh2 | 7 | 17420 | (8545, 2881, 2881) | Hourly |
| ETTm1 | 7 | 69680 | (34465, 11521, 11521) | 15 mins |
| ETTm2 | 7 | 69680 | (34465, 11521, 11521) | 15 mins |
| Weather | 321 | 52696 | (36792, 5271, 10540) | 10 mins |
| Illness | 7 | 966 | (617, 74, 170) | Weekly |
| Traffic | 862 | 17544 | (12185, 1757, 3509) | Hourly |
| Electricity | 21 | 26304 | (18317, 2633, 5261) | Hourly |

## A.2  Baselines

For TSC, following (Zhou et al., 2023), 18 conventional and SOTA baselines are included. For TSF, following (Nie et al., 2023; Tan et al., 2024; Chen et al., 2025), 8 representative LLM-based, Transfomer-based, and non-Transformer-based baselines are included. Since several baselines are used in both TSC and TSF tasks (*e.g.*, GPT4TS, Autoformer, Dlinear, *etc.*), there are 21 distinct baselines, which are described as follows.

- **GPT4TS** (Zhou et al., 2023) is a foundation model built on GPT for various of time series tasks.
- **Time-LLM** (Jin et al., 2023) implements reprogramming to align time series with language so as to leverage pre-trained LLMs.

- **CALF** (Liu et al., 2025b) is built upon LLMs by designing a cross attention and feature regularization loss to align time series with language.

- **PatchTST** (Nie et al., 2023) divides time series into subsequence-based patches, which is then modeled as tokens through Transformer encoders with channel independence strategy.

- **Flowformer** (Huang et al., 2022) introduces a linear-time attention mechanism named Flow-Attention without using specific inductive biases for time series forecasting.

- **Informer** (Zhou et al., 2021) is a Transformer-based model that designs a ProbSparse attention mechanism to reduce time complexity on long time series.

- **Transformer** (Vaswani et al., 2017) is the most traditional encoder-decoder structure which can model time series with attention mechanism.

- **Stationary** (Liu et al., 2022b) combines series stationarization and de-stationary attention to solve the over-stationarization problem in time series forecasting.

- **Refromer** (Kitaev et al., 2020) applies locality-sensitive hashing and reversible residual layers to improve the efficiency of using Transformers to model long time series.

- **Autoformer** (Wu et al., 2021) replaces the attention block of Transformer with the Auto-Correlation mechanism which can enhance both efficiency and accuracy in time series forecasting.

- **ETSformer** (Woo et al., 2022) decomposes an input time series into interpretable components with exponential smoothing attention and frequency attention for time series forecasting.

- **Pyraformer** (Liu et al., 2022a) designs a pyramidal attention module with inter-scale tree structures and intra-scale neighboring connections to capture multi-resolution temporal dependencies.

- **FEDformer** (Zhou et al., 2022) combines seasonal-trend decomposition with a frequency-enhanced Transformer to capture both global patterns and detailed structures in time series.

- **Rocket** (Dempster et al., 2020) achieves accurate time series classification by using linear classifiers with random convolutional kernels.

- **XGBoost** (Chen & Guestrin, 2016) is an efficient implementation of gradient boost decision trees for both classification and regression tasks.

- **Dlinear** (Zeng et al., 2023a) is a linear model that decomposes an input time series into seasonal component and trend component, and then models them with linear layers.

- **LightTS** (Zhang et al., 2022) is an efficient MLP-based architecture for multivariate time series forecasting by leveraging interval and continuous down-sampling to preserve temporal patterns.

- **TimesNet** (Wu et al., 2023) transforms time series into a 2D image-like representation using period-based patching, and then models the transformed time series with inception blocks.

- **TCN** (Franceschi et al., 2019) is a type of convolutional neural network that uses causal, dilated convolutions with residual connections to model the temporal dependencies in time series.

- **LSTNet** (Lai et al., 2018) integrates RNNs and CNNs to capture temporal patterns in time series.

- **LSSL** (Gu et al., 2021) is proposed based on a new parameterization for state space model to capture the long-term dependencies in time series.

## A.3 Evaluation Metrics

For TSC, following (Wu et al., 2023; Zhou et al., 2023), accuracy (in percentage) is used as the evaluation metric. For TSF, following (Nie et al., 2023; Zeng et al., 2023a; Tan et al., 2024; Chen et al., 2025), Mean Squared Error (MSE) and Mean Absolute Error (MAE) are used as the evaluation metrics. Eq. (1) defines MSE and MAE.

$$\text{MSE} = \frac{1}{D \cdot T} \sum_{d=1}^{D} \sum_{t=1}^{T} \|\hat{\mathbf{Y}}_{dt} - \mathbf{Y}_{dt}\|_2^2, \qquad \text{MAE} = \frac{1}{D \cdot T} \sum_{d=1}^{D} \sum_{t=1}^{T} \|\hat{\mathbf{Y}}_{dt} - \mathbf{Y}_{dt}\|_1 \tag{1}$$

where $\hat{\mathbf{Y}} \in \mathbb{R}^{D \times T}$ stands for the forecast at $T$ future time steps of $D$ variates, $\mathbf{Y}$ stands for the ground truth, $\|\cdot\|_2$ is $\ell_2$ norm, and $\|\cdot\|_1$ is $\ell_1$ norm.

## A.4 Experimental Settings

Following (Nie et al., 2023; Zeng et al., 2023a; Tan et al., 2024), for fair comparison, we adopt the standard evaluation protocol. In particular, the look-back window length is set to $H = 336$. The forecasting horizon is set to $T \in \{96, 192, 336, 720\}$ for all datasets except for Illness dataset. For Illness dataset, because of its limited total length of 966 time steps, shorter look-back window of $H = 104$ and forecasting horizon $T \in \{24, 36, 48, 60\}$ are employed by following (Nie et al., 2023; Zeng et al., 2023a; Tan et al., 2024). Unless otherwise noted, this configuration is applied to all of the experiments on TSF.

## A.5 Implementation Details

As described in §4, 4 pre-trained LVMs have been included in our experiments. For `ViT` and `Swin`, we use the checkpoints `ViT_B_16_Weights.IMAGENET1K_V1` and `Swin_B_Weights.IMAGENET1K_V1` respectively from `PyTorch`, which are pre-trained on $224 \times 224 \times 3$ sized images. For `MAE`, we use the checkpoint released by *Meta Research*[2], which is pre-trained on $224 \times 224 \times 3$ sized images with `ViT-Base` backbone. For `SimMIM`, we use the checkpoint released by *Microsoft*[3], which is pre-trained on $192 \times 192 \times 3$ sized images with `Swin-Base` backbone.

For TSC task, we fine-tune the LVMs using Adam optimizer with learning rate 0.0001 and batch size 32. The training runs up to 30 epochs on the training set. Early stopping is applied after 8 consecutive epochs of no improvement is observed on the validation set.

For TSF task, we use Adam optimizer with learning rate 0.0001. For ETT and Illness datasets, the batch size is set to 32. For Weather, Traffic and Electricity datasets, the batch size is set to 256. The training runs up to 20 epochs on the training set. Early stopping is applied after 3 consecutive epochs of no improvement is observed on the validation set.

All experiments are repeated three times, and the final result is obtained by taking the average. Unless otherwise noted, the above training configuration is applied to all experiments.

The experiments are conducted on NVIDIA RTX 6000 Ada Generation GPUs with 48GB memory. All implementations are based on PyTorch 2.6.0 and utilize CUDA 12.4 for training.

## A.6 Imaging Methods

In this section, we elaborate Gramian Angular Field (GAF) and Univariate Heatmap (UVH), as they are the most frequently used imaging methods in our experiments. For more details about GAF, UVH, and other imaging methods, we refer readers to (Ni et al., 2025).

**Gramian Angular Field (GAF)**. Given a univariate time series $\mathbf{x} = [x_1, ..., x_T] \in \mathbb{R}^{1 \times T}$, where $x_i$ $(1 \le i \le T)$ is the value at time step $i$, GAF applies Min-Max scaling to normalize each $x_i$ to $\hat{x}_i \in [0, 1]$.

---

[2]https://github.com/facebookresearch/mae
[3]https://github.com/microsoft/SimMIM

This normalization allows each time step to be mapped into polar coordinates with angular component $\phi_i = \arccos(\hat{x}_i)$ and radial component $r_i = i/N$, where $N$ is a constant factor.

In Gramian Sum Angular Field (GSAF), the $(i, j)$-th entry encodes the temporal correlation between time steps $i$ and $j$, which is computed as $\cos(\phi_i + \phi_j)$ and can be further expanded as following.

$$cos(\phi_i + \phi_j) = \hat{x}_i\hat{x}_j - \sqrt{1 - \hat{x}_i^2}\sqrt{1 - \hat{x}_j^2} \tag{2}$$

The resulting GAF is a matrix of size $T \times T$, with $(i, j)$-th entry defined as $\cos(\phi_i + \phi_j)$, which captures the pairwise temporal correlations among all time steps. For a multivariate time series $\mathbf{X} \in \mathbb{R}^{d \times T}$, the resulting GAF consists of $d$ individual $T \times T$ matrices.

**Univariate Heatmap (UVH)**. Given a univariate time series $\mathbf{x} \in \mathbb{R}^{1 \times T}$, UVH applies Fast Fourier Transform (FFT) to compute the Fourier coefficient of each frequency component $f_i$, where $f_i \in [1, \lfloor T/2 \rfloor]$. Then it identifies the dominant frequency $f_L$ with the largest coefficient amplitude, and sets the potential period length as $L = \lceil T/f_L \rceil$. Next, $\mathbf{x}$ is left-padded to a length-$\hat{T}$ time series $\hat{\mathbf{x}}$, where $\hat{T}$ is a multiple of $L$. The padded time series $\hat{\mathbf{x}}$ is subsequently reshaped into a 2D image of size $L \times \hat{T}/L$ by stacking its subsequences of length $L$.

**Segment length selection for UVH**. To identify the best segment length for UVH, FFT is applied on a long look-back window of 1152 time steps on all datasets except for Illness dataset, where 104 time steps is used to accommodate short time series. Table 8 summarizes the top-3 potential periods with the highest Fourier coefficients on each TSF dataset, along with the segment length $L$ used in the subsequent experiments involving UVH imaging method.

Table 8: Top-3 potential periods by FFT and segment lengths for UVH on 8 TSF datasets.

|  | ETTh1, ETTh2 | ETTm1, ETTm2 | Weather | Illness | Traffic | Electricity |
|---|---|---|---|---|---|---|
| **Top 3 Period** | {24, 576, 384} | {96, 576, 384} | {144, 72, 576} | {52, 26, 17} | {24, 12, 168} | {24, 164, 82} |
| **Segment Length L** | 24 | 96 | 144 | 52 | 24 | 24 |

# B  Full Experimental Results

## B.1  Full Results of RQ1: What type of LVM best fits the TSC (TSF) task?

The detailed performance comparison between self-supervised LVMs and supervised LVMs using the best imaging method on TSC (*i.e.*, GAF) and TSF (*i.e.*, UVH) tasks are provided in Table 9 and Table 10, respectively. For TSC, supervised and self-supervised LVMs perform comparably, while for TSF, self-supervised LVMs outperform their supervised counterparts.

Table 9: Accuracy (%) comparison between self-supervised LVMs and supervised LVMs on TSC benchmark datasets. Red numbers indicate the best performance for each dataset.

| Dataset | MAE | SimMiM | ViT | Swin |
|---|---|---|---|---|
| UWaveGestureLibrary | 85.0 | 83.1 | 88.4 | 78.9 |
| SpokenArabicDigits | 98.5 | 88.2 | 98.5 | 87.3 |
| Handwriting | 39.5 | 29.8 | 36.4 | 33.1 |
| FaceDetection | 65.4 | 57.8 | 67.4 | 50.3 |
| Average | 72.1 | 64.7 | 72.6 | 62.4 |

## B.2  Full Results of RQ2: Which imaging method best fits the TSC (TSF) task?

This section provides detailed performance comparison of 8 imaging methods, including GAF, MVH, RP, STFT, Wavelet (Wave.), Filterbank (Filter.), UVH, and Line Plot. The best-performing LVMs for TSC

Table 10: MSE and MAE Comparison between self-supervised LVMs and supervised LVMs on TSF benchmark datasets. Red numbers indicate the best performance for each forecasting horizon per dataset.

| Dataset | Model | Self-Supervised | | | | Supervised | | | |
| | | MAE | | SimMIM | | ViT | | Swin | |
| | Metrics | MSE | MAE | MSE | MAE | MSE | MAE | MSE | MAE |
|---|---|---|---|---|---|---|---|---|---|
| ETTh1 | 96 | 0.356 | 0.383 | 0.362 | 0.383 | 0.398 | 0.401 | 0.407 | 0.429 |
| | 192 | 0.395 | 0.406 | 0.407 | 0.412 | 0.439 | 0.445 | 0.442 | 0.458 |
| | 336 | 0.417 | 0.424 | 0.422 | 0.417 | 0.462 | 0.458 | 0.467 | 0.481 |
| | 720 | 0.467 | 0.463 | 0.462 | 0.455 | 0.479 | 0.491 | 0.470 | 0.497 |
| | Average | 0.409 | 0.419 | 0.413 | 0.417 | 0.445 | 0.449 | 0.447 | 0.466 |
| ETTm1 | 96 | 0.284 | 0.333 | 0.311 | 0.350 | 0.344 | 0.384 | 0.308 | 0.360 |
| | 192 | 0.328 | 0.363 | 0.335 | 0.367 | 0.414 | 0.425 | 0.350 | 0.381 |
| | 336 | 0.357 | 0.384 | 0.356 | 0.382 | 0.411 | 0.427 | 0.385 | 0.407 |
| | 720 | 0.411 | 0.417 | 0.400 | 0.413 | 0.466 | 0.451 | 0.430 | 0.437 |
| | Average | 0.345 | 0.374 | 0.351 | 0.378 | 0.409 | 0.422 | 0.368 | 0.396 |
| Weather | 96 | 0.146 | 0.191 | 0.148 | 0.196 | 0.162 | 0.219 | 0.163 | 0.216 |
| | 192 | 0.194 | 0.238 | 0.196 | 0.243 | 0.196 | 0.244 | 0.214 | 0.262 |
| | 336 | 0.243 | 0.275 | 0.244 | 0.276 | 0.250 | 0.286 | 0.270 | 0.298 |
| | 720 | 0.318 | 0.328 | 0.340 | 0.340 | 0.329 | 0.342 | 0.345 | 0.348 |
| | Average | 0.225 | 0.258 | 0.232 | 0.264 | 0.234 | 0.273 | 0.248 | 0.281 |
| Illness | 24 | 1.977 | 0.921 | 1.934 | 0.902 | 1.989 | 0.941 | 1.990 | 0.942 |
| | 36 | 1.812 | 0.872 | 1.754 | 0.825 | 2.123 | 1.002 | 2.003 | 0.951 |
| | 48 | 1.743 | 0.856 | 1.715 | 0.867 | 2.200 | 1.032 | 2.084 | 0.991 |
| | 60 | 1.816 | 0.881 | 1.673 | 0.877 | 2.404 | 1.087 | 2.128 | 1.007 |
| | Average | 1.837 | 0.883 | 1.769 | 0.868 | 2.179 | 1.016 | 2.051 | 0.973 |

Table 11: Accuracy (%) comparison of 8 imaging methods on TSC benchmark datasets. Red numbers indicate the best performance for each dataset.

| Dataset | GAF | MVH | RP | STFT | Wave. | Filter. | UVH | Lineplot |
|---|---|---|---|---|---|---|---|---|
| EthanolConcentration | 49.4 | 30.7 | 43.7 | 31.9 | 27.3 | 28.1 | 28.5 | 25.2 |
| FaceDetection | 67.4 | 68.3 | 65.5 | 61.1 | 63.9 | 64.7 | 67.7 | 50.3 |
| Handwriting | 36.4 | 30.8 | 45.1 | 28.2 | 34.0 | 22.3 | 25.8 | 15.9 |
| Heartbeat | 74.6 | 77.5 | 71.7 | 74.7 | 72.6 | 73.1 | 78.0 | 53.7 |
| Japanese Vowels | 98.3 | 97.8 | 87.8 | 94.8 | 94.9 | 97.0 | 96.4 | 65.7 |
| PEMS-SF | 84.2 | 87.2 | 80.1 | 68.5 | 84.7 | 71.2 | 88.1 | 73.4 |
| SelfRegulationSCP1 | 97.2 | 90.4 | 98.6 | 90.7 | 76.7 | 55.6 | 91.8 | 85.3 |
| SelfRegulationSCP2 | 58.8 | 53.3 | 54.4 | 52.7 | 54.4 | 52.2 | 52.8 | 44.5 |
| SpokenArabicDigits | 98.5 | 97.5 | 98.4 | 97.9 | 96.1 | 95.0 | 97.0 | 68.1 |
| UWaveGestureLibrary | 88.4 | 88.7 | 91.8 | 86.2 | 86.3 | 52.1 | 84.3 | 74.0 |
| Average | 75.3 | 72.2 | 73.7 | 68.7 | 69.1 | 61.1 | 71.0 | 55.6 |

(*i.e.*, ViT) and TSF (*i.e.*, MAE) are used. Table 11 and Table 12 summarize the results for TSC and TSF, respectively. For TSF, UVH demonstrates a clear advantage on the 4 datasets in Table 12. For TSC, all benchmark datasets are used in Table 11 because the 4 datasets outlined in §4.3 are insufficient to confidently rank the compared methods using critical difference (CD) diagram (Fig. 4). Using all datasets improves confidence and helps identify the best imaging method (*i.e.*, GAF).

In Table 12, GAF (GAF[†]) represents applying GAF with framework (c) (framework (d)) in Fig. 1. For GAF[†], we follow (Wang & Oates, 2015) to use its inverse function to recover forecasted values from the reconstructed images by the framework in Fig. 1(d). Notably, GAF[†] scales all time series values within [0, 1] by min-max normalization to compute polar coordinates during its imaging process. The normalization uses the minimum and maximum values from the look-back window, which are used to recover any predicted values. This imposes a constraint on the predicted values, *i.e.*, the predicted values must remain within the upper and lower bounds of the look-back window, which is irrational in TSF, leading to a significant limitation and

performance degradation as demonstrated in Table 12. As such, we use GAF, which outperforms GAF† in Table 12, in the CD diagram in Fig. 4(b). The key limitation of GAF (and RP, STFT, Wavelet, Filterbank, Lineplot) in TSF is because they don't preserve the original time series values in their images. Without knowing historical time series values, LVMs cannot effectively forecasting future values. MVH preserves time series values, thus is better than the above methods. However, it underperforms UVH. The limitation of MVH is its mixture of variates in the image without a principled ordering of the variates.

Table 12: MSE and MAE comparison of 8 imaging methods on TSF benchmark datasets. Red numbers indicate the best performance. GAF represents applying GAF with the framework in Fig. 1(c). GAF† represents applying GAF with the framework in Fig. 1(d).

| Imaging Method | | GAF | | GAF† | | MVH | | RP | | STFT | | Wave. | | Filter. | | UVH | | Lineplot | |
|---|---|---|---|---|---|---|---|---|---|---|---|---|---|---|---|---|---|---|---|
| Dataset | Metrics | MSE | MAE | MSE | MAE | MSE | MAE | MSE | MAE | MSE | MAE | MSE | MAE | MSE | MAE | MSE | MAE | MSE | MAE |
| ETTh1 | 96 | 0.986 | 0.783 | 1.224 | 0.850 | 0.484 | 0.471 | 0.969 | 0.771 | 0.534 | 0.533 | 0.621 | 0.582 | 0.820 | 0.684 | 0.356 | 0.383 | 0.902 | 0.751 |
| | 192 | 1.004 | 0.797 | 1.227 | 0.854 | 0.575 | 0.517 | 0.971 | 0.775 | 0.621 | 0.587 | 0.650 | 0.600 | 0.864 | 0.707 | 0.395 | 0.406 | 1.204 | 0.894 |
| | 336 | 1.038 | 0.820 | 1.214 | 0.857 | 0.623 | 0.546 | 0.989 | 0.788 | 0.602 | 0.573 | 0.681 | 0.616 | 0.827 | 0.693 | 0.417 | 0.424 | 1.223 | 0.901 |
| | 720 | 1.008 | 0.812 | 1.190 | 0.863 | 0.737 | 0.612 | 1.062 | 0.825 | 0.669 | 0.621 | 0.699 | 0.633 | 0.858 | 0.720 | 0.467 | 0.463 | 1.150 | 0.852 |
| | Average | 1.009 | 0.803 | 1.214 | 0.856 | 0.605 | 0.537 | 0.998 | 0.790 | 0.607 | 0.579 | 0.663 | 0.608 | 0.842 | 0.701 | 0.409 | 0.419 | 1.120 | 0.850 |
| ETTm1 | 96 | 0.836 | 0.729 | 0.956 | 0.676 | 0.310 | 0.352 | 0.849 | 0.719 | 0.420 | 0.470 | 0.449 | 0.490 | 0.793 | 0.648 | 0.284 | 0.333 | 0.842 | 0.735 |
| | 192 | 0.830 | 0.717 | 0.967 | 0.685 | 0.386 | 0.400 | 0.865 | 0.726 | 0.466 | 0.496 | 0.504 | 0.524 | 0.798 | 0.649 | 0.328 | 0.363 | 0.840 | 0.726 |
| | 336 | 0.853 | 0.725 | 0.988 | 0.697 | 0.393 | 0.402 | 0.872 | 0.728 | 0.506 | 0.519 | 0.532 | 0.535 | 0.883 | 0.690 | 0.357 | 0.384 | 0.841 | 0.726 |
| | 720 | 0.865 | 0.726 | 1.107 | 0.779 | 0.488 | 0.467 | 0.928 | 0.754 | 0.543 | 0.536 | 0.586 | 0.563 | 0.899 | 0.703 | 0.411 | 0.417 | 0.872 | 0.741 |
| | Average | 0.846 | 0.724 | 1.005 | 0.709 | 0.394 | 0.405 | 0.879 | 0.732 | 0.484 | 0.505 | 0.518 | 0.528 | 0.843 | 0.673 | 0.345 | 0.374 | 0.849 | 0.732 |
| Illness | 24 | 5.066 | 1.591 | 6.172 | 2.618 | 2.326 | 0.976 | 5.106 | 1.594 | 5.049 | 1.591 | 4.270 | 1.484 | 7.863 | 2.056 | 1.977 | 0.921 | 4.993 | 1.508 |
| | 36 | 5.236 | 1.628 | 5.497 | 2.627 | 2.152 | 0.919 | 5.309 | 1.629 | 5.143 | 1.598 | 4.293 | 1.487 | 8.169 | 2.122 | 1.812 | 0.872 | 5.147 | 1.593 |
| | 48 | 5.118 | 1.600 | 5.218 | 2.448 | 2.111 | 0.966 | 5.381 | 1.643 | 5.010 | 1.574 | 4.190 | 1.451 | 7.144 | 1.962 | 1.743 | 0.856 | 5.039 | 1.541 |
| | 60 | 5.349 | 1.641 | 5.299 | 2.239 | 2.118 | 0.968 | 5.586 | 1.685 | 5.164 | 1.601 | 4.045 | 1.430 | 7.193 | 1.986 | 1.816 | 0.881 | 5.235 | 1.601 |
| | Average | 5.192 | 1.615 | 5.547 | 2.483 | 2.177 | 0.957 | 5.346 | 1.638 | 5.092 | 1.591 | 4.200 | 1.463 | 7.592 | 2.032 | 1.837 | 0.883 | 5.104 | 1.561 |
| Weather | 96 | 0.581 | 0.554 | 0.961 | 0.592 | 0.153 | 0.202 | 0.647 | 0.610 | 0.202 | 0.294 | 0.224 | 0.312 | 0.515 | 0.488 | 0.146 | 0.191 | 0.588 | 0.561 |
| | 192 | 0.598 | 0.567 | 0.995 | 0.614 | 0.194 | 0.241 | 0.649 | 0.607 | 0.251 | 0.336 | 0.273 | 0.354 | 0.516 | 0.488 | 0.194 | 0.238 | 0.604 | 0.574 |
| | 336 | 0.593 | 0.558 | 1.039 | 0.637 | 0.239 | 0.275 | 0.674 | 0.619 | 0.294 | 0.364 | 0.330 | 0.388 | 0.505 | 0.484 | 0.243 | 0.275 | 0.601 | 0.568 |
| | 720 | 0.611 | 0.574 | 1.051 | 0.644 | 0.337 | 0.344 | 0.640 | 0.593 | 0.364 | 0.413 | 0.411 | 0.433 | 0.513 | 0.499 | 0.318 | 0.328 | 0.617 | 0.582 |
| | Average | 0.596 | 0.563 | 1.012 | 0.622 | 0.231 | 0.266 | 0.653 | 0.607 | 0.278 | 0.352 | 0.310 | 0.372 | 0.512 | 0.490 | 0.225 | 0.258 | 0.603 | 0.571 |

## B.3 Full Results of RQ3: Can LVMs achieve SOTA performance?

Table 13 provides the full results of the compared methods on 10 benchmark datasets for TSC. The LVM results are averaged over 3 runs. The corresponding standard deviations are reported in Table 14.

Table 13: Accuracy (%) of the compared methods on 10 TSC benchmark datasets. Red numbers indicate the the best results. # Wins is the number of times a method performs the best.

| Dataset | MAE | ViT | XGBoost | Rocket | LSTNet | LSSL | TCN | Trans. | Re. | In. | Pyra. | Auto. | Station. | FED. | ETS. | Flow. | Dlinear | LightTS | TimesNet | GPT4TS |
|---|---|---|---|---|---|---|---|---|---|---|---|---|---|---|---|---|---|---|---|---|
| EthanolConcentration | 41.4 | 49.4 | 43.7 | 45.2 | 39.9 | 31.1 | 28.9 | 32.7 | 31.9 | 31.6 | 30.8 | 31.6 | 32.7 | 31.2 | 28.1 | 33.8 | 32.6 | 29.7 | 35.7 | 34.2 |
| FaceDetection | 65.4 | 67.4 | 63.3 | 64.7 | 65.7 | 66.7 | 52.8 | 67.3 | 68.6 | 67.0 | 65.7 | 68.4 | 68.0 | 66.0 | 66.3 | 67.6 | 68.0 | 67.5 | 68.6 | 69.2 |
| Handwriting | 39.5 | 36.4 | 15.8 | 58.8 | 25.8 | 24.6 | 53.3 | 32.0 | 27.4 | 32.8 | 29.4 | 36.7 | 31.6 | 28.0 | 32.5 | 33.8 | 27.0 | 26.1 | 32.1 | 32.7 |
| Heartbeat | 86.8 | 74.6 | 73.2 | 75.6 | 77.1 | 72.7 | 75.6 | 76.1 | 77.1 | 80.5 | 75.6 | 74.6 | 73.7 | 73.7 | 71.2 | 77.6 | 75.1 | 75.1 | 78.0 | 77.2 |
| Japanese Vowels | 95.4 | 98.3 | 86.5 | 96.2 | 98.1 | 98.4 | 98.9 | 98.7 | 97.8 | 98.9 | 98.4 | 96.2 | 99.2 | 98.4 | 95.9 | 98.9 | 96.2 | 96.2 | 98.4 | 98.6 |
| PEMS-SF | 84.4 | 84.2 | 98.3 | 75.1 | 86.7 | 86.1 | 68.8 | 82.1 | 82.7 | 81.5 | 83.2 | 82.7 | 87.3 | 80.9 | 86.0 | 83.8 | 75.1 | 88.4 | 89.6 | 87.9 |
| SelfRegulationSCP1 | 95.2 | 97.2 | 84.6 | 90.8 | 84.0 | 90.8 | 84.6 | 92.2 | 90.4 | 90.1 | 88.1 | 84.0 | 89.4 | 88.7 | 89.6 | 92.5 | 87.3 | 89.8 | 91.8 | 93.2 |
| SelfRegulationSCP2 | 59.4 | 58.8 | 48.9 | 53.3 | 52.8 | 52.2 | 55.6 | 53.9 | 56.7 | 53.3 | 53.3 | 50.6 | 57.2 | 54.4 | 55.0 | 56.1 | 50.5 | 51.1 | 57.2 | 59.4 |
| SpokenArabicDigits | 98.5 | 98.5 | 69.6 | 71.2 | 100.0 | 100.0 | 95.6 | 98.4 | 97.0 | 100.0 | 99.6 | 100.0 | 100.0 | 100.0 | 100.0 | 98.8 | 81.4 | 100.0 | 99.0 | 99.2 |
| UWaveGestureLibrary | 85.0 | 88.4 | 75.9 | 94.4 | 87.8 | 85.9 | 88.4 | 85.6 | 85.6 | 85.6 | 83.4 | 85.9 | 87.5 | 85.3 | 85.0 | 86.6 | 82.1 | 80.3 | 85.3 | 88.1 |
| Average | 75.1 | 75.3 | 66.0 | 72.5 | 71.8 | 70.9 | 70.3 | 71.9 | 71.5 | 72.1 | 70.8 | 71.1 | 72.7 | 70.7 | 71.0 | 73.0 | 67.5 | 70.4 | 73.6 | 74.0 |
| # Wins | 2 | 3 | 1 | 1 | 1 | 1 | 1 | 0 | 0 | 1 | 0 | 1 | 2 | 1 | 1 | 0 | 0 | 1 | 0 | 2 |

Table 15 provides the full result of the compared methods on 8 benchmark datasets for TSF. The results of LVMs are averaged over 3 runs with standard deviations reported in Table 16.

## B.4 Full Results of RQ4: Are the pre-trained parameters in LVMs useful in time series tasks?

Table 17 and Table 18 provide the results of comparing different fine-tuning strategies on TSC and TSF tasks, respectively. In this ablation analysis, we progressively freeze the components of the Transformer blocks in LVMs (Fig. 6) with the following settings: (a) Fine-tuning all parameters; (b) Fine-tuning all parameters except the CLS and Mask tokens; (c) Fine-tuning only the MLP and normalization layers; (d) Fine-tuning

Table 14: Standard deviations of LVMs on TSC benchmark datasets.

| Dataset | MAE | ViT |
|---|---|---|
| EthanolConcentration | $41.4 \pm 0.5$ | $49.4 \pm 0.9$ |
| FaceDetection | $65.4 \pm 1.2$ | $67.4 \pm 1.5$ |
| Handwriting | $39.5 \pm 1.5$ | $36.4 \pm 1.3$ |
| Heartbeat | $86.8 \pm 2.1$ | $74.6 \pm 0.6$ |
| Japanese Vowels | $95.4 \pm 0.3$ | $98.3 \pm 0.3$ |
| PEMS-SF | $84.4 \pm 0.4$ | $84.2 \pm 0.5$ |
| SelfRegulationSCP1 | $95.2 \pm 0.6$ | $97.2 \pm 0.9$ |
| SelfRegulationSCP2 | $59.4 \pm 1.5$ | $58.8 \pm 1.3$ |
| SpokenArabicDigits | $98.5 \pm 0.5$ | $98.5 \pm 0.5$ |
| UWaveGestureLibrary | $85.0 \pm 1.7$ | $88.4 \pm 1.4$ |

Table 15: MSE and MAE evaluation of the compared methods on TSF benchmark datasets. Red (Blue) numbers are the best (second best) results on each forecasting horizon per dataset. # Wins is the number of times a method performs the best.

| Method | | MAE | | ViT | | Time-LLM | | GPT4TS | | CALF | | Dlinear | | PatchTST | | TimesNet | | FEDformer | | Autoformer | |
|---|---|---|---|---|---|---|---|---|---|---|---|---|---|---|---|---|---|---|---|---|---|
| Metrics | | MSE | MAE | MSE | MAE | MSE | MAE | MSE | MAE | MSE | MAE | MSE | MAE | MSE | MAE | MSE | MAE | MSE | MAE | MSE | MAE |
| ETTh1 | 96 | 0.356 | 0.383 | 0.398 | 0.401 | 0.376 | 0.402 | 0.370 | 0.389 | 0.370 | 0.393 | 0.375 | 0.399 | 0.370 | 0.399 | 0.384 | 0.402 | 0.376 | 0.419 | 0.449 | 0.459 |
| | 192 | 0.395 | 0.406 | 0.439 | 0.445 | 0.407 | 0.421 | 0.412 | 0.413 | 0.429 | 0.426 | 0.405 | 0.416 | 0.413 | 0.421 | 0.436 | 0.429 | 0.420 | 0.448 | 0.500 | 0.482 |
| | 336 | 0.417 | 0.424 | 0.462 | 0.458 | 0.430 | 0.438 | 0.448 | 0.431 | 0.451 | 0.440 | 0.439 | 0.443 | 0.422 | 0.436 | 0.491 | 0.469 | 0.459 | 0.465 | 0.521 | 0.496 |
| | 720 | 0.467 | 0.463 | 0.479 | 0.491 | 0.457 | 0.468 | 0.441 | 0.449 | 0.476 | 0.466 | 0.472 | 0.490 | 0.447 | 0.466 | 0.521 | 0.500 | 0.506 | 0.507 | 0.514 | 0.512 |
| ETTh2 | 96 | 0.297 | 0.341 | 0.302 | 0.355 | 0.286 | 0.346 | 0.280 | 0.335 | 0.284 | 0.336 | 0.289 | 0.353 | 0.274 | 0.336 | 0.340 | 0.374 | 0.358 | 0.397 | 0.346 | 0.388 |
| | 192 | 0.356 | 0.386 | 0.394 | 0.411 | 0.361 | 0.391 | 0.348 | 0.380 | 0.353 | 0.378 | 0.383 | 0.418 | 0.339 | 0.379 | 0.402 | 0.414 | 0.429 | 0.439 | 0.456 | 0.452 |
| | 336 | 0.371 | 0.402 | 0.423 | 0.429 | 0.390 | 0.414 | 0.380 | 0.405 | 0.361 | 0.394 | 0.448 | 0.465 | 0.329 | 0.380 | 0.452 | 0.452 | 0.496 | 0.487 | 0.482 | 0.486 |
| | 720 | 0.403 | 0.430 | 0.438 | 0.449 | 0.405 | 0.434 | 0.406 | 0.436 | 0.406 | 0.428 | 0.605 | 0.551 | 0.379 | 0.422 | 0.462 | 0.468 | 0.463 | 0.474 | 0.515 | 0.511 |
| ETTm1 | 96 | 0.284 | 0.333 | 0.344 | 0.384 | 0.291 | 0.341 | 0.300 | 0.340 | 0.323 | 0.350 | 0.299 | 0.343 | 0.290 | 0.342 | 0.338 | 0.375 | 0.379 | 0.419 | 0.505 | 0.475 |
| | 192 | 0.328 | 0.363 | 0.414 | 0.425 | 0.341 | 0.369 | 0.343 | 0.368 | 0.335 | 0.376 | 0.335 | 0.365 | 0.332 | 0.369 | 0.374 | 0.387 | 0.426 | 0.441 | 0.553 | 0.496 |
| | 336 | 0.357 | 0.384 | 0.411 | 0.427 | 0.359 | 0.379 | 0.376 | 0.386 | 0.411 | 0.401 | 0.369 | 0.386 | 0.366 | 0.392 | 0.410 | 0.411 | 0.445 | 0.459 | 0.621 | 0.537 |
| | 720 | 0.411 | 0.417 | 0.466 | 0.451 | 0.433 | 0.419 | 0.431 | 0.416 | 0.476 | 0.438 | 0.425 | 0.421 | 0.416 | 0.420 | 0.478 | 0.450 | 0.543 | 0.490 | 0.671 | 0.561 |
| ETTm2 | 96 | 0.173 | 0.258 | 0.179 | 0.265 | 0.162 | 0.248 | 0.163 | 0.249 | 0.177 | 0.255 | 0.167 | 0.269 | 0.165 | 0.255 | 0.187 | 0.267 | 0.203 | 0.287 | 0.255 | 0.339 |
| | 192 | 0.231 | 0.297 | 0.262 | 0.319 | 0.235 | 0.304 | 0.222 | 0.291 | 0.245 | 0.300 | 0.224 | 0.303 | 0.220 | 0.292 | 0.249 | 0.309 | 0.269 | 0.328 | 0.281 | 0.340 |
| | 336 | 0.282 | 0.340 | 0.346 | 0.371 | 0.280 | 0.329 | 0.273 | 0.327 | 0.309 | 0.341 | 0.281 | 0.342 | 0.274 | 0.329 | 0.321 | 0.351 | 0.325 | 0.366 | 0.339 | 0.372 |
| | 720 | 0.386 | 0.413 | 0.411 | 0.392 | 0.366 | 0.382 | 0.357 | 0.376 | 0.402 | 0.395 | 0.397 | 0.421 | 0.362 | 0.385 | 0.408 | 0.403 | 0.421 | 0.415 | 0.433 | 0.432 |
| Weather | 96 | 0.146 | 0.191 | 0.162 | 0.219 | 0.155 | 0.199 | 0.148 | 0.188 | 0.168 | 0.207 | 0.176 | 0.237 | 0.149 | 0.198 | 0.172 | 0.220 | 0.217 | 0.296 | 0.266 | 0.336 |
| | 192 | 0.194 | 0.238 | 0.196 | 0.244 | 0.223 | 0.261 | 0.192 | 0.230 | 0.216 | 0.251 | 0.220 | 0.282 | 0.194 | 0.241 | 0.219 | 0.261 | 0.276 | 0.336 | 0.307 | 0.367 |
| | 336 | 0.243 | 0.275 | 0.250 | 0.286 | 0.251 | 0.279 | 0.246 | 0.273 | 0.265 | 0.292 | 0.265 | 0.319 | 0.245 | 0.282 | 0.280 | 0.306 | 0.339 | 0.380 | 0.359 | 0.395 |
| | 720 | 0.318 | 0.328 | 0.329 | 0.342 | 0.345 | 0.342 | 0.320 | 0.328 | 0.350 | 0.345 | 0.333 | 0.362 | 0.314 | 0.334 | 0.365 | 0.359 | 0.403 | 0.428 | 0.419 | 0.428 |
| Illness | 24 | 1.977 | 0.921 | 1.989 | 0.941 | 1.792 | 0.807 | 1.869 | 0.823 | 1.460 | 0.788 | 2.215 | 1.081 | 1.319 | 0.754 | 2.317 | 0.934 | 3.228 | 1.260 | 3.483 | 1.287 |
| | 36 | 1.812 | 0.872 | 2.123 | 1.002 | 1.833 | 0.833 | 1.853 | 0.854 | 1.573 | 0.837 | 1.963 | 0.963 | 1.430 | 0.834 | 1.972 | 0.920 | 2.679 | 1.080 | 3.103 | 1.148 |
| | 48 | 1.743 | 0.856 | 2.200 | 1.032 | 2.269 | 1.012 | 1.886 | 0.855 | 1.784 | 0.890 | 2.130 | 1.024 | 1.553 | 0.815 | 2.238 | 0.940 | 2.622 | 1.078 | 2.669 | 1.085 |
| | 60 | 1.816 | 0.881 | 2.404 | 1.087 | 2.177 | 0.925 | 1.877 | 0.877 | 1.982 | 0.962 | 2.368 | 1.096 | 1.470 | 0.788 | 2.027 | 0.928 | 2.857 | 1.157 | 2.770 | 1.125 |
| Traffic | 96 | 0.346 | 0.232 | 0.403 | 0.330 | 0.392 | 0.267 | 0.396 | 0.264 | 0.416 | 0.274 | 0.410 | 0.282 | 0.360 | 0.249 | 0.593 | 0.321 | 0.587 | 0.366 | 0.613 | 0.388 |
| | 192 | 0.376 | 0.245 | 0.411 | 0.334 | 0.409 | 0.271 | 0.412 | 0.268 | 0.430 | 0.276 | 0.423 | 0.287 | 0.379 | 0.256 | 0.617 | 0.336 | 0.604 | 0.373 | 0.616 | 0.382 |
| | 336 | 0.389 | 0.252 | 0.429 | 0.335 | 0.434 | 0.296 | 0.421 | 0.273 | 0.451 | 0.286 | 0.436 | 0.296 | 0.392 | 0.264 | 0.629 | 0.336 | 0.621 | 0.383 | 0.622 | 0.337 |
| | 720 | 0.432 | 0.293 | 0.477 | 0.371 | 0.451 | 0.291 | 0.455 | 0.291 | 0.478 | 0.301 | 0.466 | 0.315 | 0.432 | 0.286 | 0.640 | 0.350 | 0.626 | 0.382 | 0.660 | 0.408 |
| Electricity | 96 | 0.127 | 0.217 | 0.152 | 0.244 | 0.137 | 0.233 | 0.141 | 0.239 | 0.147 | 0.240 | 0.140 | 0.237 | 0.129 | 0.222 | 0.168 | 0.272 | 0.193 | 0.308 | 0.201 | 0.317 |
| | 192 | 0.148 | 0.237 | 0.164 | 0.249 | 0.152 | 0.247 | 0.158 | 0.253 | 0.163 | 0.254 | 0.153 | 0.249 | 0.157 | 0.240 | 0.184 | 0.289 | 0.201 | 0.315 | 0.222 | 0.334 |
| | 336 | 0.163 | 0.253 | 0.173 | 0.275 | 0.169 | 0.267 | 0.172 | 0.266 | 0.178 | 0.270 | 0.169 | 0.267 | 0.163 | 0.259 | 0.198 | 0.300 | 0.214 | 0.329 | 0.231 | 0.338 |
| | 720 | 0.199 | 0.293 | 0.202 | 0.294 | 0.200 | 0.290 | 0.207 | 0.293 | 0.215 | 0.300 | 0.203 | 0.301 | 0.197 | 0.290 | 0.220 | 0.320 | 0.246 | 0.355 | 0.254 | 0.361 |
| # Wins | | 28 | | 0 | | 5 | | 14 | | 1 | | 0 | | 20 | | 0 | | 0 | | 0 | |

only the normalization layers; (e) Freezing all parameters; and (f) Randomly initializing an LVM and training it from scratch. From Table 17, for TSC, fully fine-tuning all parameters yields the best performance. From Table 18, for TSF, fine-tuning only the normalization layers leads to better performance than other settings.

Table 16: Standard deviations of LVMs on TSF benchmark datasets.

| Method | | MAE | | ViT | |
|---|---|---|---|---|---|
| Metrics | | MSE | MAE | MSE | MAE |
| ETTh1 | 96 | 0.356 ± 0.001 | 0.383 ± 0.005 | 0.398 ± 0.011 | 0.401 ± 0.012 |
| | 192 | 0.395 ± 0.001 | 0.406 ± 0.001 | 0.439 ± 0.005 | 0.445 ± 0.003 |
| | 336 | 0.417 ± 0.001 | 0.424 ± 0.001 | 0.462 ± 0.004 | 0.458 ± 0.004 |
| | 720 | 0.467 ± 0.012 | 0.463 ± 0.010 | 0.479 ± 0.011 | 0.491 ± 0.008 |
| ETTh2 | 96 | 0.297 ± 0.000 | 0.341 ± 0.004 | 0.302 ± 0.001 | 0.355 ± 0.000 |
| | 192 | 0.356 ± 0.005 | 0.386 ± 0.011 | 0.394 ± 0.001 | 0.411 ± 0.001 |
| | 336 | 0.371 ± 0.003 | 0.402 ± 0.004 | 0.423 ± 0.003 | 0.429 ± 0.001 |
| | 720 | 0.403 ± 0.001 | 0.430 ± 0.005 | 0.438 ± 0.005 | 0.449 ± 0.002 |
| ETTm1 | 96 | 0.284 ± 0.003 | 0.333 ± 0.004 | 0.344 ± 0.001 | 0.384 ± 0.002 |
| | 192 | 0.328 ± 0.001 | 0.363 ± 0.002 | 0.414 ± 0.003 | 0.425 ± 0.003 |
| | 336 | 0.357 ± 0.001 | 0.384 ± 0.001 | 0.411 ± 0.002 | 0.427 ± 0.007 |
| | 720 | 0.411 ± 0.002 | 0.417 ± 0.001 | 0.466 ± 0.003 | 0.451 ± 0.002 |
| ETTm2 | 96 | 0.173 ± 0.005 | 0.258 ± 0.004 | 0.179 ± 0.003 | 0.265 ± 0.004 |
| | 192 | 0.231 ± 0.004 | 0.297 ± 0.003 | 0.262 ± 0.002 | 0.319 ± 0.001 |
| | 336 | 0.282 ± 0.001 | 0.340 ± 0.004 | 0.346 ± 0.001 | 0.371 ± 0.003 |
| | 720 | 0.386 ± 0.002 | 0.413 ± 0.003 | 0.411 ± 0.002 | 0.392 ± 0.004 |
| Weather | 96 | 0.146 ± 0.000 | 0.191 ± 0.002 | 0.162 ± 0.001 | 0.219 ± 0.003 |
| | 192 | 0.194 ± 0.001 | 0.238 ± 0.002 | 0.196 ± 0.002 | 0.244 ± 0.003 |
| | 336 | 0.243 ± 0.000 | 0.275 ± 0.001 | 0.250 ± 0.001 | 0.286 ± 0.000 |
| | 720 | 0.318 ± 0.001 | 0.328 ± 0.001 | 0.329 ± 0.002 | 0.342 ± 0.002 |
| Illness | 24 | 1.977 ± 0.017 | 0.921 ± 0.003 | 1.989 ± 0.011 | 0.941 ± 0.004 |
| | 36 | 1.812 ± 0.014 | 0.872 ± 0.009 | 2.123 ± 0.006 | 1.002 ± 0.003 |
| | 48 | 1.743 ± 0.029 | 0.856 ± 0.012 | 2.200 ± 0.009 | 1.032 ± 0.005 |
| | 60 | 1.816 ± 0.022 | 0.881 ± 0.008 | 2.404 ± 0.018 | 1.087 ± 0.011 |
| Traffic | 96 | 0.346 ± 0.004 | 0.232 ± 0.003 | 0.403 ± 0.003 | 0.330 ± 0.002 |
| | 192 | 0.376 ± 0.006 | 0.245 ± 0.002 | 0.411 ± 0.001 | 0.334 ± 0.000 |
| | 336 | 0.389 ± 0.004 | 0.252 ± 0.003 | 0.429 ± 0.002 | 0.335 ± 0.005 |
| | 720 | 0.432 ± 0.002 | 0.293 ± 0.005 | 0.477 ± 0.004 | 0.371 ± 0.002 |
| Electricity | 96 | 0.127 ± 0.001 | 0.217 ± 0.000 | 0.152 ± 0.001 | 0.244 ± 0.001 |
| | 192 | 0.148 ± 0.004 | 0.237 ± 0.000 | 0.164 ± 0.003 | 0.249 ± 0.001 |
| | 336 | 0.163 ± 0.001 | 0.253 ± 0.002 | 0.173 ± 0.002 | 0.275 ± 0.003 |
| | 720 | 0.199 ± 0.002 | 0.293 ± 0.001 | 0.202 ± 0.001 | 0.294 ± 0.003 |

Table 17: Accuracy (%) comparison of different fine-tuning strategies on TSC benchmark datasets. Red numbers indicate the best performance for each dataset.

| Dataset | (a) | (b) | (c) | (d) | (e) | (f) |
|---|---|---|---|---|---|---|
| UWaveGestureLibrary | 88.4 | 87.5 | 88.7 | 81.6 | 84.0 | 73.4 |
| SpokenArabicDigits | 98.5 | 98.2 | 98.4 | 98.0 | 98.5 | 97.0 |
| Handwriting | 36.4 | 35.2 | 35.5 | 28.5 | 27.8 | 24.3 |
| FaceDetection | 67.4 | 66.3 | 67.1 | 65.2 | 66.7 | 65.0 |

Table 18: MSE and MAE comparison of different fine-tuning strategies on TSF benchmark datasets. Red numbers indicate the best performance for each dataset.

| Fine-tuning Strategy | | (a) | | (b) | | (c) | | (d) | | (e) | | (f) | |
|---|---|---|---|---|---|---|---|---|---|---|---|---|---|
| Dataset | Metrics | MSE | MAE | MSE | MAE | MSE | MAE | MSE | MAE | MSE | MAE | MSE | MAE |
| ETTh1 | 96 | 0.512 | 0.448 | 0.481 | 0.435 | 0.477 | 0.418 | 0.356 | 0.383 | 0.426 | 0.397 | 0.412 | 0.431 |
| | 192 | 0.511 | 0.453 | 0.520 | 0.455 | 0.526 | 0.456 | 0.395 | 0.406 | 0.448 | 0.417 | 0.462 | 0.462 |
| | 336 | 0.610 | 0.512 | 0.537 | 0.484 | 0.584 | 0.497 | 0.417 | 0.424 | 0.478 | 0.439 | 0.489 | 0.479 |
| | 720 | 0.598 | 0.523 | 0.581 | 0.526 | 0.539 | 0.493 | 0.467 | 0.463 | 0.454 | 0.453 | 0.536 | 0.514 |
| | Average | 0.558 | 0.484 | 0.530 | 0.475 | 0.532 | 0.466 | 0.409 | 0.419 | 0.452 | 0.427 | 0.475 | 0.472 |
| ETTm1 | 96 | 0.303 | 0.334 | 0.320 | 0.348 | 0.306 | 0.338 | 0.284 | 0.333 | 0.394 | 0.370 | 0.323 | 0.367 |
| | 192 | 0.385 | 0.385 | 0.389 | 0.385 | 0.385 | 0.378 | 0.328 | 0.363 | 0.404 | 0.381 | 0.344 | 0.383 |
| | 336 | 0.409 | 0.403 | 0.419 | 0.407 | 0.420 | 0.402 | 0.357 | 0.384 | 0.421 | 0.398 | 0.375 | 0.403 |
| | 720 | 0.500 | 0.461 | 0.503 | 0.461 | 0.474 | 0.444 | 0.411 | 0.417 | 0.462 | 0.426 | 0.446 | 0.445 |
| | Average | 0.399 | 0.396 | 0.408 | 0.400 | 0.396 | 0.391 | 0.345 | 0.374 | 0.420 | 0.394 | 0.372 | 0.400 |
| Illness | 24 | 1.888 | 0.818 | 1.683 | 0.789 | 2.043 | 0.818 | 1.977 | 0.921 | 2.227 | 0.971 | 1.719 | 0.799 |
| | 36 | 1.542 | 0.781 | 1.632 | 0.801 | 1.573 | 0.775 | 1.812 | 0.872 | 2.023 | 0.932 | 1.541 | 0.753 |
| | 48 | 1.682 | 0.829 | 1.839 | 0.845 | 1.548 | 0.783 | 1.743 | 0.856 | 1.947 | 0.920 | 1.687 | 0.817 |
| | 60 | 2.012 | 0.859 | 1.977 | 0.921 | 1.783 | 0.860 | 1.816 | 0.881 | 1.952 | 0.939 | 1.944 | 0.880 |
| | Average | 1.781 | 0.822 | 1.783 | 0.839 | 1.737 | 0.809 | 1.837 | 0.883 | 2.037 | 0.941 | 1.723 | 0.812 |
| Weather | 96 | 0.172 | 0.213 | 0.174 | 0.213 | 0.171 | 0.208 | 0.146 | 0.191 | 0.274 | 0.280 | 0.154 | 0.201 |
| | 192 | 0.225 | 0.259 | 0.233 | 0.263 | 0.225 | 0.256 | 0.194 | 0.238 | 0.284 | 0.294 | 0.199 | 0.245 |
| | 336 | 0.298 | 0.302 | 0.296 | 0.304 | 0.293 | 0.303 | 0.243 | 0.275 | 0.311 | 0.316 | 0.265 | 0.292 |
| | 720 | 0.397 | 0.363 | 0.397 | 0.364 | 0.367 | 0.361 | 0.318 | 0.328 | 0.364 | 0.354 | 0.344 | 0.350 |
| | Average | 0.273 | 0.284 | 0.275 | 0.286 | 0.264 | 0.282 | 0.225 | 0.258 | 0.308 | 0.311 | 0.241 | 0.272 |

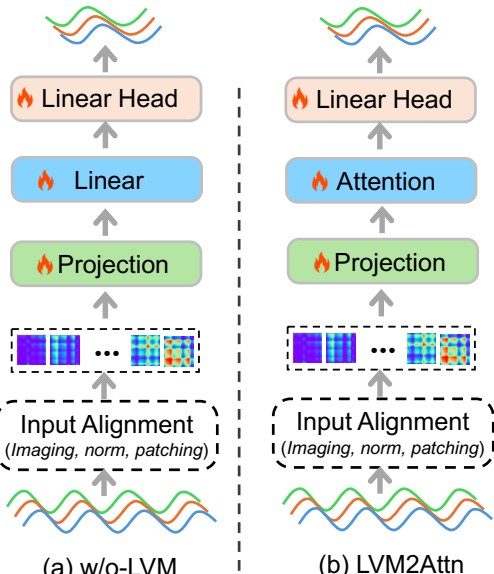

Figure 12: An illustration of the ablation models. (a) is the model W/O-LVM, which replaces the Transformer blocks in LVMs with a linear layer. (b) is the model LVM2ATTN, which replaces the Transformer blocks in LVMs with a single multi-head attention layer.

### B.5    Full Results of RQ5: How useful are LVM architectures?

Table 19 and Table 20 provide the results of comparing LVM architecture and two ablation models, W/O-LVM and LVM2Attn, on TSC and TSF tasks, respectively. Fig. 12 illustrates the ablation models. Both models keep the projection layer in LVM encoder. The model W/O-LVM replaces the Transformer blocks with a linear layer. The model LVM2Attn replaces the Transformer blocks with a single multi-head self-attention layer. Other components including input alignment and the linear head remain unchanged. In this comparison, all models are trained from scratch without using pre-trained parameters. From Table 19 and Table 20, without pre-trained knowledge, LVMs perform on par with W/O-LVM and LVM2Attn on both TSC and TSF tasks. However, as demonstrated in Table 17 and Table 18, with pre-training parameters, LVMs outperform both ablation models.

Table 19: Accuracy (%) comparison between LVM architecture and ablation models on TSC benchmark datasets. Red numbers indicate the best performance for each dataset.

| Dataset | LVM | W/O-LVM | LVM2Attn |
|---|---|---|---|
| UWaveGestureLibrary | 73.4 | 78.6 | 80.1 |
| SpokenArabicDigits | 97.0 | 96.4 | 96.5 |
| Handwriting | 24.3 | 22.4 | 20.7 |
| FaceDetection | 65.0 | 64.1 | 66.2 |

Table 20: MSE and MAE comparison between LVM architecture and ablation models on TSF benchmark datasets. Red numbers indicate the best performance for each dataset.

| Model | | LVM | | W/O-LVM | | LVM2Attn | |
|---|---|---|---|---|---|---|---|
| Dataset | Metrics | MSE | MAE | MSE | MAE | MSE | MAE |
| ETTh1 | 96 | 0.412 | 0.431 | 0.392 | 0.410 | 0.391 | 0.417 |
| | 192 | 0.462 | 0.462 | 0.418 | 0.426 | 0.414 | 0.435 |
| | 336 | 0.489 | 0.479 | 0.441 | 0.443 | 0.438 | 0.452 |
| | 720 | 0.536 | 0.514 | 0.441 | 0.465 | 0.469 | 0.485 |
| | Average | 0.475 | 0.472 | 0.423 | 0.436 | 0.428 | 0.447 |
| ETTm1 | 96 | 0.323 | 0.367 | 0.322 | 0.364 | 0.298 | 0.354 |
| | 192 | 0.344 | 0.383 | 0.353 | 0.381 | 0.338 | 0.380 |
| | 336 | 0.375 | 0.403 | 0.388 | 0.401 | 0.376 | 0.401 |
| | 720 | 0.446 | 0.445 | 0.440 | 0.432 | 0.416 | 0.427 |
| | Average | 0.372 | 0.400 | 0.376 | 0.395 | 0.357 | 0.391 |
| Illness | 24 | 1.719 | 0.799 | 2.280 | 1.034 | 1.990 | 0.909 |
| | 36 | 1.541 | 0.753 | 2.224 | 1.018 | 1.913 | 0.899 |
| | 48 | 1.687 | 0.817 | 2.296 | 1.039 | 2.105 | 0.964 |
| | 60 | 1.944 | 0.880 | 2.364 | 1.052 | 2.423 | 1.033 |
| | Average | 1.723 | 0.812 | 2.291 | 1.036 | 2.108 | 0.951 |
| Weather | 96 | 0.154 | 0.201 | 0.188 | 0.243 | 0.184 | 0.240 |
| | 192 | 0.199 | 0.245 | 0.226 | 0.273 | 0.226 | 0.271 |
| | 336 | 0.265 | 0.292 | 0.270 | 0.302 | 0.271 | 0.303 |
| | 720 | 0.344 | 0.350 | 0.336 | 0.347 | 0.335 | 0.346 |
| | Average | 0.241 | 0.272 | 0.255 | 0.291 | 0.254 | 0.290 |

### B.6    Full Results of RQ6: Can LVMs capture temporal dependencies in time series?

Four types of perturbations, **Sf-All**, **Sf-Half**, **Ex-Half** and **Masking**, are applied to time series to compare the performance drop of LVMs, W/O-LVM, and LVM2Attn on both TSC and TSF tasks. Table 21 and Table 22 summarize the results. As can be seen, LVMs are more vulnerable to temporal perturbations than the ablation models, suggesting that they can capture temporal dependencies in time series.

Table 21: Comparison of accuracy (%) and performance drop (%) between LVMs and the ablation models under temporal perturbations on the TSC benchmark datasets. Red numbers indicate the largest performance drop for each dataset.

| Model | | LVMs | | | | w/o-LVM | | | | LVM2ATTN | | | |
|---|---|---|---|---|---|---|---|---|---|---|---|---|---|
| Dataset | Perturbation | Shuffle All | Shuffle Half | Ex-half | Masking | Shuffle All | Shuffle Half | Ex-half | Masking | Shuffle All | Shuffle Half | Ex-half | Masking |
| UWaveGestureLibrary | Accuracy(%) | 17.1 | 56.2 | 35.9 | 62.8 | 17.1 | 73.4 | 0.9 | 79.4 | 10.9 | 73.1 | 0.9 | 79.3 |
| | Performance Drop | 80.7% | 36.4% | 59.4% | 29.0% | 78.2% | 6.6% | 98.8% | -1.0% | 86.4% | 8.7% | 98.9% | 1.0% |
| SpokenArabicDigits | Accuracy(%) | 15.1 | 68.8 | 9.9 | 57.3 | 48.5 | 84.4 | 17.2 | 93.4 | 47.7 | 85.3 | 17.2 | 93.0 |
| | Performance Drop | 84.7% | 30.2% | 89.9% | 41.8% | 49.7% | 12.4% | 82.2% | 3.1% | 50.6% | 11.6% | 82.2% | 3.6% |
| Handwriting | Accuracy(%) | 3.1 | 4.9 | 1.1 | 16.0 | 4.1 | 5.7 | 3.7 | 17.4 | 2.1 | 3.4 | 2.7 | 16.5 |
| | Performance Drop | 91.5% | 86.5% | 97.0% | 56.0% | 81.7% | 74.6% | 83.5% | 22.3% | 89.9% | 83.6% | 87.0% | 20.3% |
| FaceDetection | Accuracy(%) | 47.7 | 61.1 | 61.2 | 62.4 | 51.7 | 57.2 | 49.5 | 64.9 | 51.4 | 58.7 | 49.9 | 64.4 |
| | Performance Drop | 29.2% | 9.3% | 9.2% | 7.4% | 19.3% | 10.8% | 22.8% | -1.2% | 22.4% | 11.3% | 24.6% | 2.7% |

Table 22: Comparison of performance drop (%) between the LVM and the ablation models under temporal perturbations on the TSF benchmark datasets. Red numbers indicate the largest performance drop among for each dataset.

| Model | | LVMs | | | | | | | | w/o-LVM | | | | | | | | LVM2Attn | | | | | | | |
|---|---|---|---|---|---|---|---|---|---|---|---|---|---|---|---|---|---|---|---|---|---|---|---|---|---|
| Perturbation | | Sf-All | | Sf-Half | | Ex-half | | Masking | | Sf-All | | Sf-Half | | Ex-half | | Masking | | Sf-All | | Sf-Half | | Ex-half | | Masking | |
| Dataset | Metrics | MSE | MAE | MSE | MAE | MSE | MAE | MSE | MAE | MSE | MAE | MSE | MAE | MSE | MAE | MSE | MAE | MSE | MAE | MSE | MAE | MSE | MAE | MSE | MAE |
| ETTh1 | 96 | 0.747 | 0.588 | 0.369 | 0.393 | 0.457 | 0.437 | 0.551 | 0.534 | 0.746 | 0.582 | 0.437 | 0.438 | 0.483 | 0.460 | 0.608 | 0.559 | 0.741 | 0.589 | 0.442 | 0.449 | 0.456 | 0.448 | 0.577 | 0.554 |
| | 192 | 0.734 | 0.584 | 0.443 | 0.446 | 0.462 | 0.444 | 0.578 | 0.550 | 0.751 | 0.590 | 0.487 | 0.468 | 0.481 | 0.461 | 0.621 | 0.567 | 0.776 | 0.622 | 0.515 | 0.502 | 0.458 | 0.452 | 0.608 | 0.578 |
| | 336 | 0.733 | 0.595 | 0.486 | 0.469 | 0.453 | 0.442 | 0.612 | 0.577 | 0.736 | 0.591 | 0.503 | 0.479 | 0.470 | 0.460 | 0.625 | 0.574 | 0.769 | 0.626 | 0.537 | 0.504 | 0.468 | 0.467 | 0.643 | 0.602 |
| | 720 | 0.765 | 0.631 | 0.587 | 0.549 | 0.480 | 0.476 | 0.664 | 0.582 | 0.740 | 0.613 | 0.509 | 0.507 | 0.472 | 0.484 | 0.635 | 0.597 | 0.779 | 0.660 | 0.554 | 0.518 | 0.479 | 0.491 | 0.669 | 0.632 |
| | Avg. Drop | 83.8% | 43.5% | 14.5% | 10.4% | 14.2% | 7.6% | 47.5% | 34.2% | 76.2% | 36.4% | 14.4% | 8.5% | 13.0% | 7.1% | 47.3% | 31.8% | 79.7% | 39.7% | 19.5% | 10.3% | 9.1% | 4.0% | 46.0% | 32.3% |
| ETTm1 | 96 | 0.732 | 0.561 | 0.441 | 0.440 | 1.127 | 0.691 | 0.504 | 0.508 | 0.731 | 0.561 | 0.441 | 0.430 | 0.929 | 0.629 | 0.567 | 0.538 | 0.779 | 0.611 | 0.442 | 0.447 | 0.895 | 0.625 | 0.577 | 0.554 |
| | 192 | 0.721 | 0.562 | 0.512 | 0.462 | 1.146 | 0.704 | 0.534 | 0.519 | 0.731 | 0.563 | 0.463 | 0.444 | 0.894 | 0.618 | 0.589 | 0.547 | 0.768 | 0.585 | 0.436 | 0.442 | 0.929 | 0.639 | 0.525 | 0.526 |
| | 336 | 0.736 | 0.568 | 0.522 | 0.492 | 1.163 | 0.724 | 0.552 | 0.533 | 0.731 | 0.568 | 0.485 | 0.457 | 0.895 | 0.622 | 0.586 | 0.547 | 0.730 | 0.569 | 0.464 | 0.454 | 0.873 | 0.622 | 0.552 | 0.537 |
| | 720 | 0.780 | 0.587 | 0.556 | 0.526 | 1.221 | 0.745 | 0.570 | 0.547 | 0.753 | 0.582 | 0.529 | 0.484 | 0.919 | 0.636 | 0.616 | 0.562 | 0.772 | 0.721 | 0.743 | 0.585 | 0.939 | 0.656 | 0.771 | 0.628 |
| | Avg. Drop | 118.4% | 53.0% | 48.2% | 28.4% | 242.3% | 92.2% | 58.4% | 41.4% | 98.4% | 44.6% | 28.3% | 15.2% | 145.3% | 59.3% | 58.5% | 39.5% | 117.1% | 59.3% | 44.8% | 23.2% | 158.3% | 63.4% | 70.3% | 44.0% |
| Illness | 24 | 4.794 | 1.578 | 2.426 | 1.064 | 2.465 | 1.045 | 4.169 | 1.386 | 5.220 | 1.674 | 3.091 | 1.251 | 2.529 | 1.098 | 4.394 | 1.507 | 4.712 | 1.613 | 3.449 | 1.287 | 2.942 | 1.219 | 4.768 | 1.572 |
| | 36 | 4.719 | 1.572 | 2.240 | 1.006 | 2.256 | 0.995 | 4.128 | 1.372 | 4.966 | 1.634 | 3.181 | 1.281 | 2.505 | 1.095 | 4.388 | 1.486 | 4.240 | 1.523 | 3.517 | 1.132 | 2.648 | 1.136 | 4.683 | 1.533 |
| | 48 | 4.665 | 1.561 | 2.108 | 0.964 | 2.157 | 0.974 | 4.113 | 1.373 | 4.685 | 1.583 | 3.240 | 1.294 | 2.487 | 1.089 | 4.428 | 1.480 | 4.179 | 1.515 | 3.615 | 1.359 | 2.463 | 1.070 | 4.689 | 1.540 |
| | 60 | 5.094 | 1.622 | 2.138 | 0.962 | 2.161 | 0.999 | 4.374 | 1.422 | 4.947 | 1.632 | 3.464 | 1.335 | 2.648 | 1.129 | 4.574 | 1.521 | 4.349 | 1.523 | 3.597 | 1.352 | 2.629 | 1.099 | 4.940 | 1.578 |
| | Avg. Drop | 162.8% | 79.5% | 21.3% | 13.2% | 23.0% | 13.7% | 128.9% | 57.4% | 116.4% | 57.5% | 41.6% | 24.6% | 11.0% | 6.5% | 94.1% | 44.7% | 109.1% | 62.9% | 69.3% | 34.8% | 27.9% | 19.5% | 127.8% | 64.0% |
| Weather | 96 | 0.258 | 0.316 | 0.162 | 0.211 | 0.329 | 0.351 | 0.278 | 0.371 | 0.261 | 0.312 | 0.189 | 0.246 | 0.295 | 0.329 | 0.290 | 0.379 | 0.261 | 0.313 | 0.189 | 0.246 | 0.298 | 0.331 | 0.290 | 0.380 |
| | 192 | 0.283 | 0.329 | 0.206 | 0.249 | 0.336 | 0.354 | 0.296 | 0.371 | 0.291 | 0.331 | 0.230 | 0.276 | 0.320 | 0.342 | 0.315 | 0.392 | 0.288 | 0.329 | 0.232 | 0.280 | 0.315 | 0.339 | 0.315 | 0.393 |
| | 336 | 0.318 | 0.349 | 0.260 | 0.291 | 0.358 | 0.365 | 0.327 | 0.398 | 0.320 | 0.349 | 0.286 | 0.319 | 0.334 | 0.351 | 0.346 | 0.411 | 0.319 | 0.347 | 0.278 | 0.312 | 0.339 | 0.353 | 0.339 | 0.405 |
| | 720 | 0.396 | 0.411 | 0.363 | 0.357 | 0.391 | 0.388 | 0.384 | 0.439 | 0.370 | 0.380 | 0.341 | 0.354 | 0.382 | 0.382 | 0.376 | 0.428 | 0.371 | 0.381 | 0.341 | 0.353 | 0.387 | 0.385 | 0.376 | 0.428 |
| | Avg. Drop | 44.50% | 38.97% | 9.57% | 7.44% | 67.20% | 45.88% | 49.58% | 57.17% | 24.06% | 18.68% | 2.43% | 2.49% | 33.98% | 21.74% | 33.42% | 39.75% | 24.43% | 19.11% | 2.44% | 2.70% | 35.49% | 22.70% | 33.58% | 40.18% |

### B.7 Full Results of RQ7: What are the computational costs of LVMs?

Fig. 13 presents the accuracy and inference efficiency comparison between LVMs and the two best-performing baselines on TSC task. Fig. 14 (Fig. 15) presents the MSE (MAE) and inference efficiency comparisons between LVMs and the two best-performing baselines on TSF task. In general, LVMs yield improved performance with higher costs of inference time.

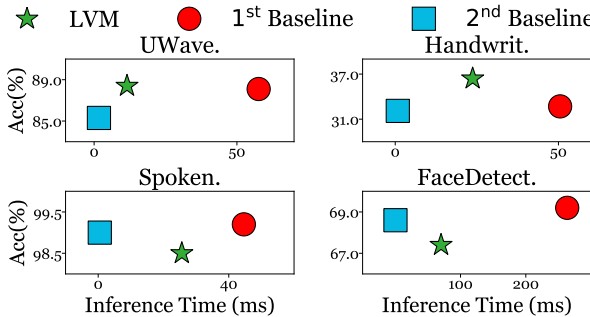

Figure 13: Accuracy *vs.* inference time of the compared methods on TSC benchmark datasets. Green marker stands for LVM, Red marker stands for `GPT4TS` and Blue marker stands for `TimesNet`.

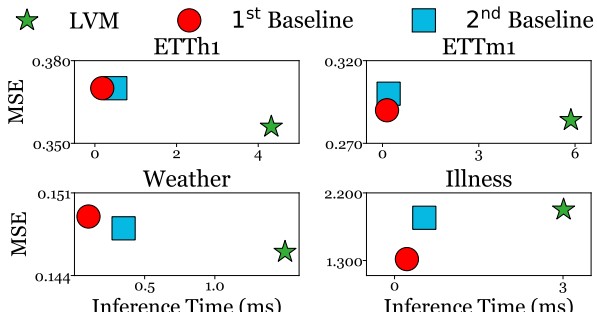

Figure 14: MSE *vs.* inference time of the compared methods on TSF benchmark datasets. Green marker stands for LVM, Red marker stands for `PatchTST` and Blue marker stands for `GPT4TS`.

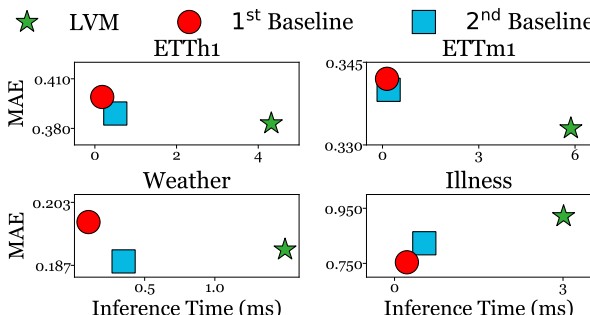

Figure 15: MAE *vs.* inference time of the compared methods on TSF benchmark datasets. Green marker stands for LVM, Red marker stands for `PatchTST` and Blue marker stands for `GPT4TS`.

### B.8 Full Results of RQ8: Which component of LVMs contributes more to TSF?

Table 23 provides the detailed results on MSE and MAE of the two ablations, **Enc w/o Dec** and **Dec w/o Enc**, of self-supervised LVMs on TSF benchmark datasets. From Table 23, **Enc w/o Dec** shows inferior performance to **Dec w/o Enc**, highlighting the importance of the pre-trained decoders of LVMs in TSF.

Table 23: MSE and MAE comparison of self-supervised LVMs with either the pre-trained encoder (**Dec w/o Enc**) or decoder (**Enc w/o Dec**) excluded on TSF benchmark datasets.

| Model | | MAE | | | | | | SimMIM | | | | | |
|---|---|---|---|---|---|---|---|---|---|---|---|---|---|
| | | Pre-trained | | Enc w/o Dec | | Dec w/o Enc | | Pre-trained | | Enc w/o Dec | | Dec w/o Enc | |
| Dataset | Metrics | MSE | MAE | MSE | MAE | MSE | MAE | MSE | MAE | MSE | MAE | MSE | MAE |
| ETTh1 | 96 | 0.356 | 0.383 | 0.420 | 0.423 | 0.396 | 0.401 | 0.362 | 0.383 | 0.466 | 0.426 | 0.412 | 0.418 |
| | 192 | 0.395 | 0.406 | 0.445 | 0.446 | 0.399 | 0.414 | 0.407 | 0.412 | 0.496 | 0.455 | 0.457 | 0.446 |
| | 336 | 0.417 | 0.424 | 0.489 | 0.484 | 0.441 | 0.433 | 0.422 | 0.417 | 0.499 | 0.474 | 0.581 | 0.520 |
| | 720 | 0.467 | 0.463 | 0.582 | 0.543 | 0.426 | 0.451 | 0.462 | 0.455 | 0.505 | 0.481 | 0.564 | 0.526 |
| | Average | 0.409 | 0.419 | 0.484 | 0.474 | 0.416 | 0.425 | 0.413 | 0.417 | 0.492 | 0.459 | 0.504 | 0.478 |
| ETTm1 | 96 | 0.284 | 0.333 | 0.324 | 0.363 | 0.295 | 0.335 | 0.311 | 0.350 | 0.320 | 0.347 | 0.299 | 0.348 |
| | 192 | 0.328 | 0.363 | 0.361 | 0.387 | 0.330 | 0.364 | 0.335 | 0.367 | 0.377 | 0.377 | 0.344 | 0.378 |
| | 336 | 0.357 | 0.384 | 0.398 | 0.414 | 0.365 | 0.388 | 0.356 | 0.382 | 0.411 | 0.401 | 0.403 | 0.419 |
| | 720 | 0.411 | 0.417 | 0.446 | 0.440 | 0.409 | 0.416 | 0.400 | 0.413 | 0.468 | 0.442 | 0.431 | 0.433 |
| | Average | 0.345 | 0.374 | 0.382 | 0.401 | 0.350 | 0.376 | 0.351 | 0.378 | 0.394 | 0.392 | 0.369 | 0.395 |
| Illness | 24 | 1.977 | 0.921 | 1.946 | 0.842 | 1.774 | 0.841 | 1.934 | 0.902 | 2.314 | 0.944 | 2.034 | 0.899 |
| | 36 | 1.812 | 0.872 | 1.981 | 0.895 | 1.918 | 0.876 | 1.754 | 0.825 | 2.434 | 1.045 | 2.198 | 0.983 |
| | 48 | 1.743 | 0.856 | 1.967 | 0.855 | 2.061 | 0.943 | 1.715 | 0.867 | 2.008 | 0.869 | 2.209 | 0.960 |
| | 60 | 1.816 | 0.881 | 1.956 | 0.858 | 1.969 | 0.950 | 1.673 | 0.877 | 1.979 | 0.865 | 2.275 | 0.997 |
| | Average | 1.837 | 0.883 | 1.963 | 0.863 | 1.931 | 0.903 | 1.769 | 0.868 | 2.184 | 0.931 | 2.179 | 0.960 |
| Weather | 96 | 0.146 | 0.191 | 0.168 | 0.210 | 0.155 | 0.201 | 0.148 | 0.196 | 0.166 | 0.208 | 0.150 | 0.200 |
| | 192 | 0.194 | 0.238 | 0.237 | 0.263 | 0.209 | 0.248 | 0.196 | 0.243 | 0.228 | 0.257 | 0.199 | 0.246 |
| | 336 | 0.243 | 0.275 | 0.299 | 0.306 | 0.274 | 0.298 | 0.244 | 0.276 | 0.294 | 0.297 | 0.251 | 0.284 |
| | 720 | 0.318 | 0.328 | 0.396 | 0.372 | 0.378 | 0.361 | 0.340 | 0.340 | 0.382 | 0.357 | 0.343 | 0.342 |
| | Average | 0.225 | 0.258 | 0.275 | 0.288 | 0.254 | 0.277 | 0.232 | 0.264 | 0.268 | 0.280 | 0.236 | 0.268 |

### B.9 Full Results of RQ9: Does the period-based UVH imaging method introduce any bias in TSF?

Fig. 16 provides the forecasting performance of an LVM (*i.e.*, MAE) in terms of evaluation metrics MAE *w.r.t.* varying segment length from $\frac{1}{6}L$ to $\frac{12}{6}L$, where $L$ is the period of the time series. The LVM generally achieves the best performance when segment length is a multiple of the period, *i.e.* $L$ or $2L$, which is caused by the inductive bias as discussed in RQ9 in §4.4.

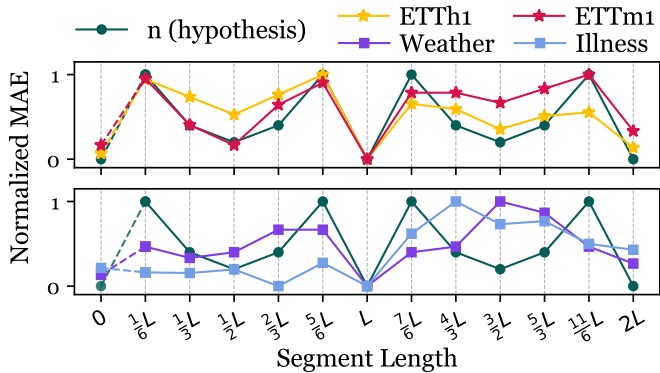

Figure 16: TSF performance (MAE) of MAE *w.r.t.* varying segment length used in the UVH imaging method across datasets. "$n$" (green) estimates TSF difficulty, as described in RQ9.

## B.10   Evaluation of LVMs on Datasets with Weak Periodicity

In this section, we compare the best LVM forecaster (*i.e.*, `MAE`) with the best non-LVM forecaster (*i.e.*, `PatchTST` according to Table 2) on datasets with weak periodicity, including Weather, Illness and two additional datasets – Exchange (Lai et al., 2018) and Solar (Liu et al., 2024a). Fig. 17 visualizes sample time series from these datasets, which demonstrate weak periodicity. Table 24 summarizes the results, from which we can observe that `MAE` does not show advantage over the non-LVM baseline on these datasets. This confirms our analysis of LVMs' bias toward periodicity in RQ9 (§4.4). The findings also call for diversifying benchmark datasets for TSF task.

Table 24: MSE and MAE comparison of LVM forcaster and non-LVM forecaster on TSF datasets with weak periodicity.

| Model | Dataset | Weather | | Illness | | Exchange | | Solar | | # Wins |
|---|---|---|---|---|---|---|---|---|---|---|
| | Metric | MSE | MAE | MSE | MAE | MSE | MAE | MSE | MAE | |
| MAE | 96 | **0.146** | **0.191** | 1.977 | 0.921 | 0.099 | 0.224 | 0.190 | **0.245** | |
| | 192 | 0.194 | **0.238** | 1.812 | 0.872 | **0.199** | 0.321 | 0.206 | **0.257** | |
| | 336 | **0.243** | **0.275** | 1.743 | 0.856 | 0.383 | 0.453 | 0.214 | **0.265** | 12 |
| | 720 | 0.318 | **0.328** | 1.816 | 0.881 | 0.937 | 0.729 | 0.235 | 0.299 | |
| | Avearge | **0.225** | **0.258** | 1.837 | 0.883 | 0.405 | 0.432 | 0.211 | 0.267 | |
| PatchTST | 96 | 0.149 | 0.198 | **1.319** | **0.754** | **0.092** | **0.213** | **0.185** | 0.251 | |
| | 192 | **0.193** | 0.241 | **1.430** | **0.834** | 0.207 | **0.235** | **0.194** | 0.263 | |
| | 336 | 0.245 | 0.282 | **1.553** | **0.815** | **0.376** | **0.451** | **0.213** | 0.274 | 29 |
| | 720 | **0.314** | 0.334 | **1.470** | **0.788** | **0.858** | **0.692** | **0.213** | **0.275** | |
| | Avearge | **0.225** | 0.264 | **1.443** | **0.798** | **0.383** | **0.398** | **0.201** | **0.266** | |

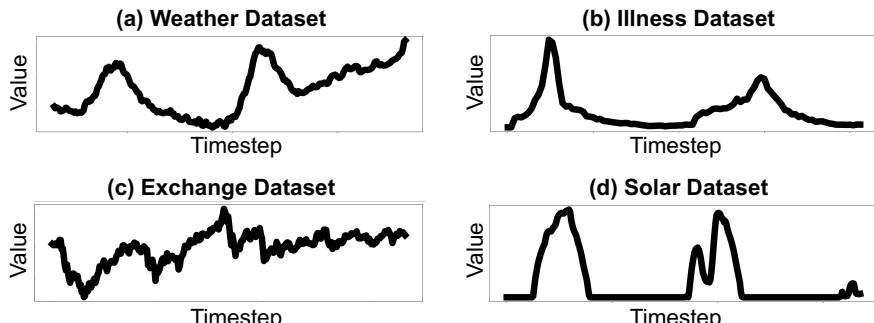

Figure 17: An illustration of random samples in a look-back window (length=336) from (a) Weather dataset; (b) Illness dataset; (c) Exchange dataset; and (d) Solar dataset, which have weak periodicity.

## B.11   Full Results of RQ10: Can LVMs make effective use of look-back windows?

Table 25 presents the MSE and MAE performance of LVMs across varying look-back window lengths, ranging from 48 to 2304. As discussed in RQ10, LVMs exhibits limited ability in fully leveraging the information of look-back windows when the window length exceeds approximately 1000 time steps. The Illness dataset is omitted in Table 25 because its time series are of short lengths, with only 966 time steps in total.

## B.12   Comparison with SOTA Diffusion-based LVM

Table 26 and Table 27 present comparisons with a SOTA diffusion-based LVM, `LaVin-DiT` (Wang et al., 2025b), on both TSC and TSF tasks. For each task, the best foundational LVM is used to compare with `LaVin-DiT`. The results does not demonstrate a significant advantage of `LaVin-DiT` over the foundational LVMs used in this work for time series analysis.

Table 25: The MSE and MAE performance of LVMs across different look-back window lengths on TSF benchmark datasets.

| Look-back Window | | 48 | | 96 | | 192 | | 336 | | 720 | | 1152 | | 1728 | | 2304 | |
|---|---|---|---|---|---|---|---|---|---|---|---|---|---|---|---|---|
| Dataset | Metrics | MSE | MAE | MSE | MAE | MSE | MAE | MSE | MAE | MSE | MAE | MSE | MAE | MSE | MAE | MSE | MAE |
| ETTh1 | 96 | 0.376 | 0.395 | 0.373 | 0.390 | 0.364 | 0.383 | 0.356 | 0.383 | 0.347 | 0.375 | 0.347 | 0.376 | 0.344 | 0.376 | 0.373 | 0.402 |
| | 192 | 0.440 | 0.431 | 0.424 | 0.418 | 0.411 | 0.412 | 0.395 | 0.406 | 0.385 | 0.405 | 0.384 | 0.402 | 0.391 | 0.408 | 0.399 | 0.417 |
| | 336 | 0.474 | 0.450 | 0.471 | 0.445 | 0.456 | 0.437 | 0.417 | 0.424 | 0.408 | 0.418 | 0.410 | 0.418 | 0.395 | 0.413 | 0.408 | 0.423 |
| | 720 | 0.485 | 0.477 | 0.482 | 0.471 | 0.469 | 0.465 | 0.467 | 0.463 | 0.468 | 0.460 | 0.432 | 0.440 | 0.425 | 0.442 | 0.424 | 0.442 |
| | Average | 0.444 | 0.438 | 0.438 | 0.431 | 0.425 | 0.424 | 0.409 | 0.419 | 0.402 | 0.415 | 0.393 | 0.409 | 0.389 | 0.410 | 0.401 | 0.421 |
| ETTm1 | 96 | 0.443 | 0.413 | 0.316 | 0.353 | 0.304 | 0.345 | 0.284 | 0.333 | 0.279 | 0.324 | 0.280 | 0.332 | 0.277 | 0.322 | 0.285 | 0.326 |
| | 192 | 0.476 | 0.431 | 0.373 | 0.390 | 0.333 | 0.365 | 0.328 | 0.363 | 0.322 | 0.358 | 0.321 | 0.361 | 0.321 | 0.355 | 0.318 | 0.350 |
| | 336 | 0.512 | 0.457 | 0.385 | 0.400 | 0.370 | 0.390 | 0.357 | 0.384 | 0.356 | 0.381 | 0.362 | 0.383 | 0.352 | 0.378 | 0.346 | 0.374 |
| | 720 | 0.574 | 0.489 | 0.449 | 0.438 | 0.426 | 0.429 | 0.411 | 0.417 | 0.411 | 0.414 | 0.399 | 0.413 | 0.411 | 0.414 | 0.407 | 0.416 |
| | Average | 0.501 | 0.448 | 0.381 | 0.395 | 0.358 | 0.382 | 0.345 | 0.374 | 0.342 | 0.369 | 0.341 | 0.372 | 0.340 | 0.367 | 0.339 | 0.367 |
| Weather | 96 | 0.200 | 0.237 | 0.167 | 0.209 | 0.152 | 0.196 | 0.146 | 0.191 | 0.142 | 0.188 | 0.144 | 0.194 | 0.143 | 0.193 | 0.141 | 0.195 |
| | 192 | 0.236 | 0.267 | 0.212 | 0.249 | 0.200 | 0.240 | 0.194 | 0.238 | 0.188 | 0.235 | 0.189 | 0.237 | 0.195 | 0.242 | 0.200 | 0.253 |
| | 336 | 0.293 | 0.307 | 0.268 | 0.290 | 0.254 | 0.280 | 0.243 | 0.275 | 0.247 | 0.281 | 0.242 | 0.279 | 0.272 | 0.302 | 0.278 | 0.307 |
| | 720 | 0.370 | 0.358 | 0.346 | 0.340 | 0.330 | 0.333 | 0.318 | 0.328 | 0.334 | 0.341 | 0.332 | 0.339 | 0.344 | 0.349 | 0.372 | 0.357 |
| | Average | 0.275 | 0.292 | 0.248 | 0.272 | 0.234 | 0.262 | 0.225 | 0.258 | 0.228 | 0.261 | 0.227 | 0.262 | 0.239 | 0.272 | 0.248 | 0.278 |

Table 26: The performance of SOTA diffusion-based LVM on TSC benchmark datasets

| Dataset | ViT | LaVin-DiT |
|---|---|---|
| UWaveGestureLibrary | 88.4 | 84.2 |
| SpokenArabicDigits | 98.5 | 97.9 |
| Handwriting | 36.4 | 36.7 |
| FaceDetection | 67.4 | 67.0 |
| Average | 72.6 | 71.5 |

Table 27: The performance of SOTA diffusion-based LVM on TSF benchmark datasets

| | MAE | | LaVin-DiT | |
|---|---|---|---|---|
| Dataset | MSE | MAE | MSE | MAE |
| ETTh1 | 0.409 | 0.419 | 0.403 | 0.416 |
| ETTm1 | 0.345 | 0.374 | 0.349 | 0.377 |
| Weather | 0.225 | 0.258 | 0.231 | 0.259 |
| Illness | 1.837 | 0.883 | 1.733 | 0.859 |

## C  Proof of Lemma 4.1

In this section, we provide the proof for Lemma 4.1.

*Proof.* Suppose $\mathbf{x}$ is exactly periodic, $x_t = x_{t+\alpha \cdot L}$ holds when $\alpha \in \mathbb{N}^+$ and $L$ is the period. The smallest number of segments $n$ before any segment reoccurs, *i.e.*, $\mathbf{x}_t = \mathbf{x}_{t+n \cdot (i/k)L}$, indicates $n \cdot (i/k) \in \mathbb{N}^+$. Hence, the proof of Lemma 4.1 is equivalent to prove $n = \frac{k}{\mathrm{GCD}(i,k)}$ as the smallest natural number such that $k$ divides $n \cdot i$, denoted as $k \mid n \cdot i$.

Set $d = \mathrm{GCD}(i,k)$ as the greatest common divisor of $i$ and $k$. The following is based on the definition of greated common divisor:

$$i = d \cdot i' \tag{3}$$

$$k = d \cdot k' \tag{4}$$

$$\mathrm{GCD}(i',k') = 1 \tag{5}$$

where $i', k' \in \mathbb{N}^+$. As $k$ divides $n \cdot i$, we have

$$k \mid n \cdot i \Rightarrow d \cdot k' \mid d \cdot n \cdot i'$$
$$\Rightarrow k' \mid n \cdot i'$$
$$\Rightarrow k' \mid n \tag{6}$$

The first step in Eq. (6) is expanded with Eq. (3) and Eq. (4). The second step cancels the common factor $d$ out from both sides with the divisibility relation unchanged. The last step follows Eq. (5). To satisfy Eq. (6), the smallest $n$ is $n = k'$. Finally, expand $k'$ with Eq. (4), we reach

$$n = k' = \frac{k}{d} = \frac{k}{\text{GCD}(i, k)}$$

$\square$

# D  Visualization Results

## D.1  Visualization of GAF on TSC Task

To have a sense about what temporal patterns can be recognized by LVMs for TSC, we visualize the images of GAF method on the Handwriting and UWaveGestureLibrary datasets in Fig. 18 and Fig. 19, respectively. The examples are randomly sampled from five different classes on both datasets. From Fig. 18 and Fig. 19, we can observe clear visual patterns that distinguish the GAF images from different classes, which highlights the effectiveness of GAF as a way to encode time series for TSC.

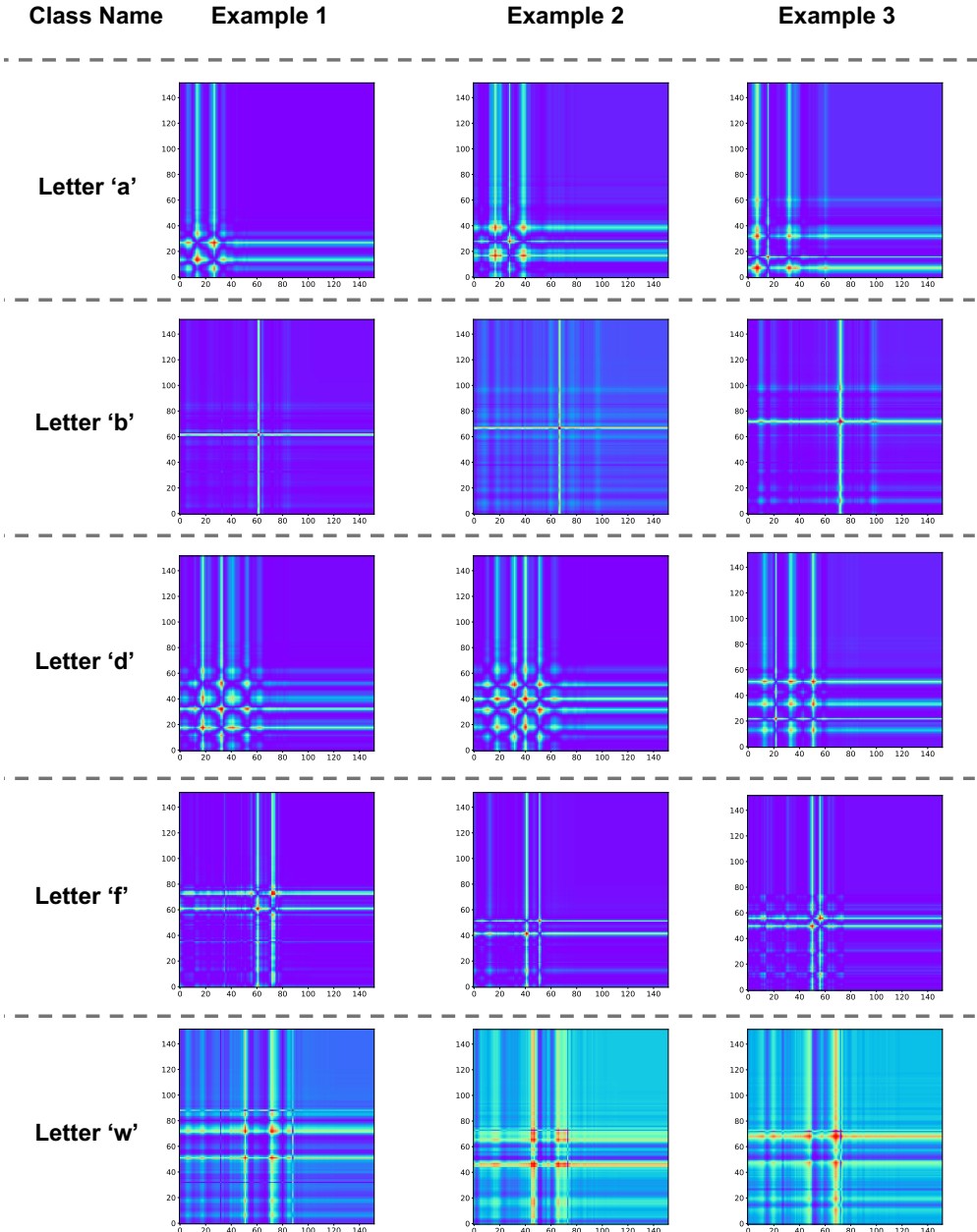

Figure 18: Examples of GAF images on the first channel of MTS with 152 time steps randomly drawn from the five classes in the Handwriting dataset.

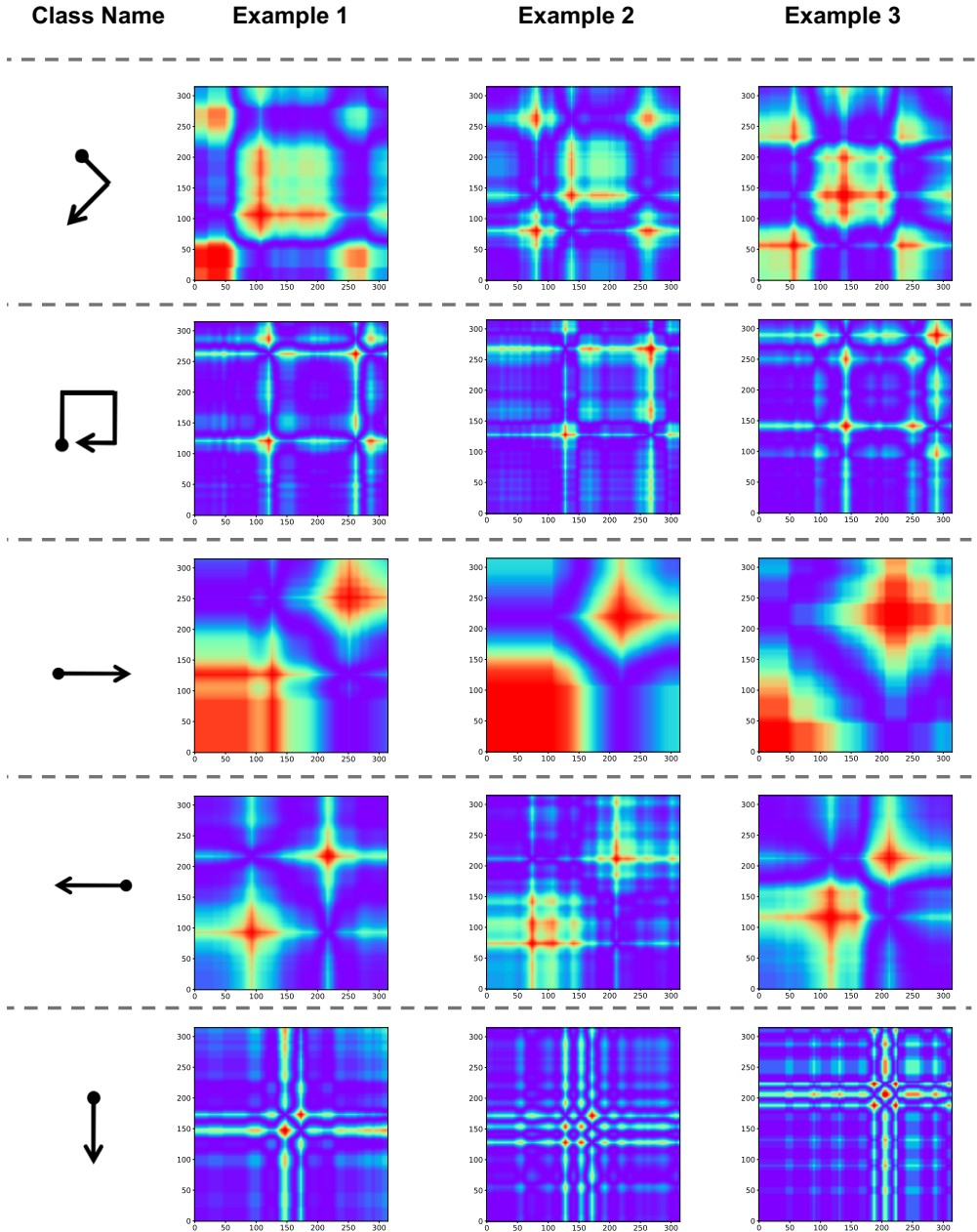

Figure 19: Examples of GAF images on the first channel of MTS with 336 time steps randomly drawn from the five classes in the UWaveGestureLibrary dataset.

### D.2 Illustration of The Inductive Bias of LVMs during TSF

As discussed in RQ9, the imaging method UVH can induce an inductive bias to LVMs in TSF toward "forecasting periods" by rendering them to combine the past segments to infer future. To illustrate this, Fig. 20 and Fig. 21 visualize two random examples with varying segment lengths from one period (24 time steps) to two periods (48 time steps) from ETTh1 and Traffic datasets. The blue lines represent the time series in the look-back windows, the red lines represent the ground truth in the forecasting horizon, and the green lines represent the forecasted time series by LVMs. The results demonstrate that LVMs perform best when the segment length aligns with the period of the time series, while the performance degrades when the segment length shifts from the period. This indicates the inductive bias toward periodicity when using UVH imaging method for TSF.

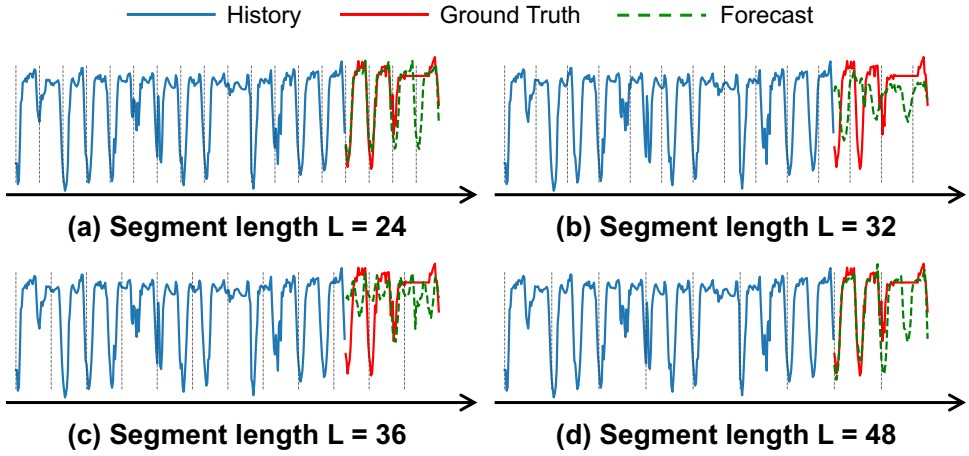

Figure 20: An illustration of the inductive bias toward periodicity during TSF on a random example from the ETTh1 dataset (period is 24 time steps). From (a) to (d), the segment length vary within {24, 32, 36, 48}.

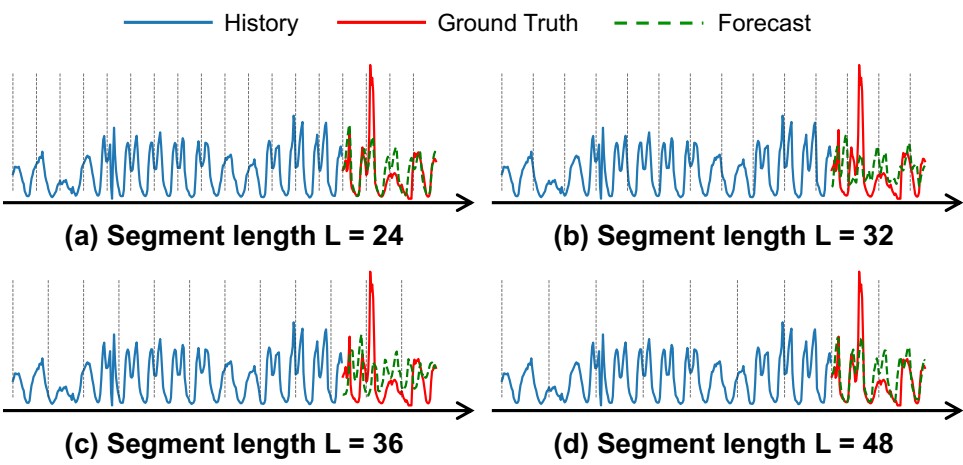

Figure 21: An illustration of the inductive bias toward periodicity during TSF on a random example from the Traffic dataset (period is 24 time steps). From (a) to (d), the segment length vary within {24, 32, 36, 48}.

## D.3 Analysis of The Decoders of LVM-based Forecasters

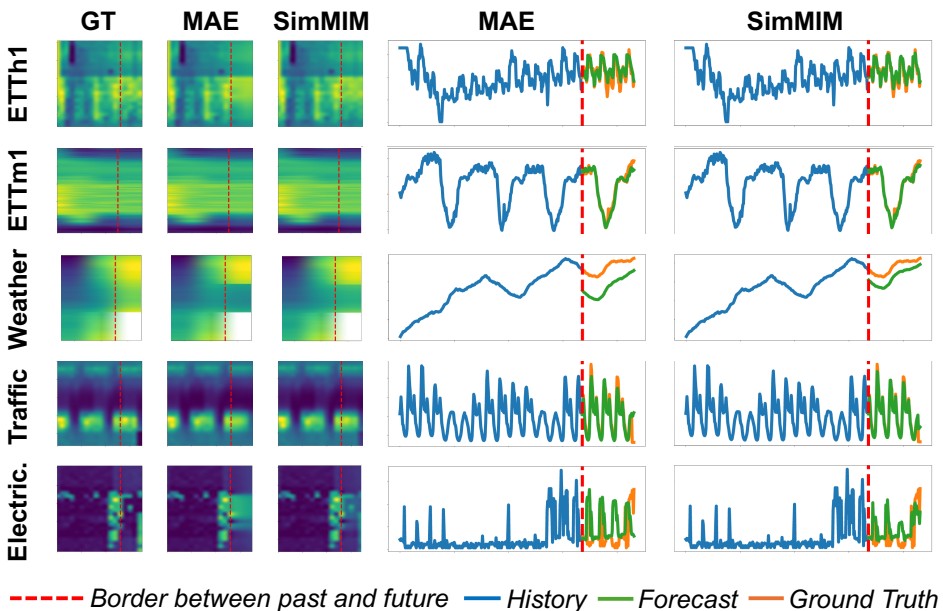

Figure 22: An illustration of the reconstructed images of `MAE`'s decoder and `SimMIM`'s decoder on five TSF benchmark datasets. The red dashed lines separate look-back windows and forecasting horizons. The first column shows the ground truth forecasts. The second and third columns show `MAE`'s forecasts and `SimMIM`'s forecasts, respectively. The fourth and fifth columns show the recovered time series of `MAE`'s forecasts and `SimMIM`'s forecasts, respectively.

In this section, we analyze the reconstruction abilities of `MAE`'s decoder and `SimMIM`'s decoder. The two encoder-decoder LVMs have different decoders – `MAE` has a 8-layer Transformer decoder and `SimMIM` has a single-layer linear decoder. In Fig. 22, we visualize the reconstructed images of both `MAE` and `SimMIM`, and compare the reconstructions with the ground truths on random samples from 5 TSF benchmark datasets – ETTh1, ETTm1, Weather, Traffic, and Electricity. In Fig. 22, the red dashed lines separate the look-back windows and the forecasting horizons. In addition to the UVH images, we visualize the recovered time series from the images. From Fig. 22, we obverse the reconstruction ability of the decoders of the LVMs on the masked areas (which corresponds to the forecasting horizons), which tend to be smooth across columns. This confirms our analysis of the bias toward periodicity in RQ9 (§4.4). It also illustrates the working mechanism of the decoders of the LVMs in the TSF task.

## E Further Discussions

**Related Works on Multimodal Methods for Time Series Analysis**. Among the recent VLM-based methods for time series analysis Wimmer & Rekabsaz (2023); Prithyani et al. (2024); Zhuang et al. (2024); Zhong et al. (2025); Shen et al. (2025a), the most relevant includes `Time-VLM` (Zhong et al., 2025) and `DMMV` (Shen et al., 2025a). `Time-VLM` builds a forecaster on `ViLT` (Kim et al., 2021) to encode numerical and visual views, along with contextual texts. While integrating rich information with a large model, `Time-VLM` demonstrates promising results in TSF. However, its fusion strategy closely follows the `ViLT` backbone and lacks time-series-specific design, leading to potentially suboptimal performance. `DMMV` integrates LVMs and numerical forecasters (*e.g.*, Transformer) in an adaptive decomposition framework to form a multimodal architecture, which aims to mitigate the bias of LVM-based forecasters. A recent survey (Jiang et al., 2025) provides a structured discussion on multimodal methods for time series analysis. However, existing works lack fundamental understandings of the effectiveness of sole LVMs in time series analysis. The goal of this work is to fill this gap.

**Further Analysis for RQ8.** The ablation study in Fig. 8 validates the importance of the decoders of self-supervised LVMs in TSF. Our understanding is that this is because the decoders of the LVMs aim to reconstruct pixel values, while the encoders aim to extract general-purpose features. For a forecasting task, reconstructing pixel values align better with forecasting the numerical values in a time series, thus plays a more important role than the encoders.

**Further Analysis for RQ10.** From Fig. 11, the ineffective use of look-back windows may stem from image transformation: fixed-size image in pre-trained LVMs has a pixel limit and may constrain the information captured from excessively long time series. Additionally, the pixels in the fixed-size image are not fully utilized by the SOTA LVM-based forecasters. In their imaging setup, each column of pixels (i.e., $256 \times 1$) represents an interpolated period with $P$ timesteps (*e.g.*, $P = 24$ for ETTh1), less than 256 timesteps. Also, following the masking strategy in `VisionTS` (Chen et al., 2025), an alignment constant $c = 0.4$ is applied, which means that over 60% of the pixels are masked. As a result, only about $P \times 0.4 \times 256$ time steps can be effectively used in one image. For example, when $P = 24$, only around 2,400 timesteps can be encoded, which is significantly fewer than the full $256 \times 256$ pixels. Thus the amount of effective pixels is influenced by $P$ and $c$. Other factors such noises and repeated patterns (*i.e.*, redundancy) may further reduce the amount of effective pixels that can inform forecasting. This raises a future direction on how to better utilize image pixels for time series forecasting.

# F Analysis of Different Imaging Methods

## F.1 Effectiveness of Different Imaging Methods

Different imaging techniques may lead to different successes of LVMs in time series analysis. This relates to (1) the property of each imaging technique; and (2) the effectiveness of the property when using LVMs.

First, the key property of each imaging technique can be deterministically identified from its definitions as specified in (Ni et al., 2025). In Fig. 23 (and Fig. 1(a)), we've empirically visualized these methods. In the following, we summarize the key properties of each imaging technique according to the formal (mathematical) definitions in (Ni et al., 2025). We'd like to refer interested readers to (Ni et al., 2025) for the detailed definitions.

- **Line Plot** (*e.g.*, Fig. 23(a)) uses a line to represent a time series in a 2D image. Its use of pixels is ineffective compared to other imaging techniques because most of the pixels are used to represent the white background.

- **MVH** (*e.g.*, Fig. 23(b)) is the only method that directly visualizes all variates in an MTS in a single heatmap image. However, there is no principled way to determine the order of variates on the y-axis. Different orders may lead to different cross-variate patterns learned by an LVM.

- **UVH** (*e.g.*, Fig. 23(c)) visualizes stacked segments of a time series as a heatmap image. As such, it may lead to a bias toward periodicity when the segments are periods.

- **GAF** (*e.g.*, Fig. 23(d)) is a square $T$-by-$T$ image ($T$ is the length of look-back window). It is good at capturing the temporal correlations and cyclical patterns in a time series.

- **STFT** (*e.g.*, Fig. 23(e)) visualizes the time-frequency space of a time series. However, it uses a fixed-size sliding window which cannot fit varying frequencies in a time series.

- **Wavelet** (*e.g.*, Fig. 23(f)) visualizes the time-frequency space of a time series using wavelets. It addresses STFT's limitation but needs a proper choice of wavelet function.

- **Filterbank** (*e.g.*, Fig. 23(g)) resembles STFT but is associated with several processing filters on a Mel-scale that fits audio signals thus is more often used for imaging audio signals.

- **RP** (*e.g.*, Fig. 23(h)) is a square image and its size is tunable by its hyperparameters. It is good at capturing periodic patterns but is a black-white image thus may lose some fine-grained information.

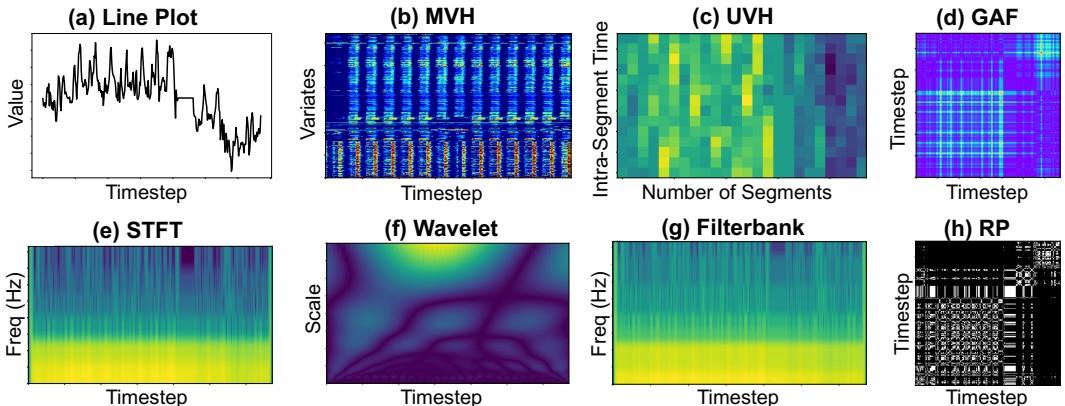

Figure 23: An illustration of different methods for imaging a time series sample (length=336) from the *Electricity* benchmark dataset.

Among these methods, only UVH and MVH images encode the raw (or normalized) time series values, while other methods visualize different transformations of time series values. This will lead to their different performance in a forecasting (numerical-level) task. Additionally, STFT, Wavelet, and Filterbank produce spectograms. Their difference lies in their different ways of encoding frequencies in a time series.

Second, our experiments for comparing the 8 imaging methods provide insights on the effectiveness of these methods when using LVMs. For classification, the averaged results in Table 11 (the bottom row) suggest that Line Plot are Filterbank are less effective than other methods. This is because of Line Plot's ineffective use of pixels and Filterbank's filter processing that is aligned with high-frequency audio signals. For the other 6 methods, despite their different levels of effectiveness, they all achieve reasonable averaged accuracy ($\sim$70%), suggesting their effectiveness in capturing distinguishable patterns (*e.g.*, GAF, RP, UVH capture periodic patterns; STFT, Wavelet capture time-frequency patterns; MVH captures overall trend patterns).

For forecasting, the results in Table 12 suggest that UVH and MVH are better than other methods. This is because forecasting task requires predicting future time series values. Among the 8 methods, only UVH and MVH encode historical time series values, thus they are more effective than other transformation methods in a numerical-level task, *i.e.*, forecasting. However, MVH underperforms UVH. The limitation of MVH is its mixture of variates in the image without a principled ordering of the variates, which may confuse cross-variate dependency learning.

## F.2 Potential Biases of Using Imaging Methods

Transforming time series to an image may introduce some biases. We summarize the key positive bias (pros) and negative bias (cons) of different imaging methods as following. The methods that do not preserve raw time series values (*i.e.*, LinePlot, STFT, Wavelet, Filterbank, GAF, RP) do not fit reconstruction-based LVMs for time series forecasting, leading to their inferior performance in Table 12. From the perspective of LVMs, the key advantage of using imaged time series is that it enables transferring the visual knowledge encoded in LVMs acquired by large-scale image-based pre-training to time series tasks. The key limitations, according to our analysis in RQ1, RQ8 and RQ10, includes (1) ineffective use of window-based local attention; (2) less effective use of encoder; and (3) fixed number of usable pixels (which caps the length of look-back windows).

- **Line Plot** (*e.g.*, Fig. 23(a)): (1) Pros: compact representation of long time series; (2) Cons: pixels are not fully used (large white background pixels).

- **MVH** (*e.g.*, Fig. 23(b)): (1) Pros: direct representation of all variates in a multivariate time series; (2) Cons: random ordering of variates may confuse correlation learning.

- **UVH** (*e.g.*, Fig. 23(c)): (1) Pros: encode periodic patterns of time series; (2) Cons: break continuity across different periods.

- **GAF** (*e.g.*, Fig. 23(d)): (1) Pros: encode temporal correlations of pairwise time steps; (2) Cons: lose raw time series values, use more pixels for longer time series ($O(T^2)$ complexity)).

- **STFT** (*e.g.*, Fig. 23(e)): (1) Pros: encode time-frequency space; (2) Cons: lose raw time series values (which is important in numerical-level tasks).

- **Wavelet** (*e.g.*, Fig. 23(f)): (1) Pros: encode time-frequency space while fitting varying frequencies; (2) Cons: lose raw time series values.

- **Filterbank** (*e.g.*, Fig. 23(g)): (1) Pros: encode time-frequency space while fitting high-frequency signals: (2) Cons: lose raw time series values.

- **RP** (*e.g.*, Fig. 23(h)): (1) Pros: encode cyclic patterns of time series; (2) Cons: lose raw time series values, back-white image may lose fine-grained information.

## G   Comparing LVM-based Forecaster with TSFMs

Table 28: MSE and MAE comparison between `MAE` and a TSFM on TSF benchmark datasets.

| Model | Dataset | ETTh1 | | ETTh2 | | ETTm1 | | ETTm2 | | Weather | | Illness | | Traffic | | Electricity | | # Wins |
|---|---|---|---|---|---|---|---|---|---|---|---|---|---|---|---|---|---|---|
| | Metric | MSE | MAE | MSE | MAE | MSE | MAE | MSE | MAE | MSE | MAE | MSE | MAE | MSE | MAE | MSE | MAE | |
| MAE | 96 | 0.356 | 0.383 | 0.297 | 0.341 | 0.284 | 0.333 | 0.173 | 0.258 | 0.146 | 0.191 | 1.977 | 0.921 | 0.346 | 0.232 | 0.127 | 0.217 | |
| | 192 | 0.395 | 0.406 | 0.356 | 0.386 | 0.328 | 0.363 | 0.231 | 0.297 | 0.194 | 0.238 | 1.812 | 0.872 | 0.376 | 0.245 | 0.148 | 0.237 | |
| | 336 | 0.417 | 0.424 | 0.371 | 0.402 | 0.357 | 0.384 | 0.282 | 0.340 | 0.243 | 0.275 | 1.743 | 0.856 | 0.389 | 0.252 | 0.163 | 0.253 | 51 |
| | 720 | 0.467 | 0.463 | 0.403 | 0.430 | 0.411 | 0.417 | 0.386 | 0.413 | 0.318 | 0.328 | 1.816 | 0.881 | 0.432 | 0.293 | 0.199 | 0.293 | |
| | Average | 0.409 | 0.419 | 0.357 | 0.390 | 0.345 | 0.374 | 0.268 | 0.327 | 0.225 | 0.258 | 1.837 | 0.883 | 0.386 | 0.256 | 0.159 | 0.250 | |
| LightGTS | 96 | 0.346 | 0.382 | 0.271 | 0.369 | 0.291 | 0.338 | 0.165 | 0.242 | 0.146 | 0.173 | 3.001 | 1.205 | 0.406 | 0.316 | 0.144 | 0.237 | |
| | 192 | 0.398 | 0.419 | 0.338 | 0.378 | 0.362 | 0.378 | 0.229 | 0.289 | 0.197 | 0.219 | 2.977 | 1.231 | 0.423 | 0.318 | 0.160 | 0.257 | |
| | 336 | 0.403 | 0.423 | 0.359 | 0.401 | 0.416 | 0.419 | 0.295 | 0.346 | 0.243 | 0.257 | 3.110 | 1.295 | 0.439 | 0.326 | 0.183 | 0.283 | 33 |
| | 720 | 0.427 | 0.450 | 0.386 | 0.420 | 0.599 | 0.489 | 0.381 | 0.397 | 0.310 | 0.301 | 3.008 | 1.289 | 0.487 | 0.354 | 0.260 | 0.339 | |
| | Average | 0.394 | 0.419 | 0.339 | 0.392 | 0.417 | 0.406 | 0.268 | 0.319 | 0.224 | 0.238 | 3.024 | 1.255 | 0.439 | 0.329 | 0.187 | 0.279 | |

In this section, we compare the LVM-based forecaster, *i.e.* `MAE`, with a time series foundation model (TSFM), *i.e.*, `LightGTS` (Wang et al., 2025a), which was pre-trained on large-scale time series datasets while is more parameter-efficient than some existing TSFMs such as `Chronos` (Ansari et al., 2025) and `Moirai` (Morid et al., 2023). Meanwhile, `LightGTS` was demonstrated to outperform TSFMs with larger sizes on TSF benchmark datasets in (Wang et al., 2025a). Table 28 summarizes the comparison on the 8 TSF benchmark datasets. From Table 28, we observe LVM's strong potential in transferring cross-modal knowledge, which appears to be more useful than the intra-modal knowledge in `LightGTS` (obtained by pre-training on time series datasets). We think this advantage attributes to the much larger size of the available pre-training image datasets than that of the pre-training time series datasets collected by existing methods. The results also suggest a potential research direction of exploring the integration of cross-modal knowledge and intra-modal knowledge for enhancing performance.

