# OpenReview forum: "From Images to Signals: Are Large Vision Models Useful for Time Series Analysis?"
_TMLR — Under review for TMLR_

### Review · Reviewer_kmW6 · 2026-06-14

**Summary Of Contributions:**

This paper benchmarks LVMs on two core time-series tasks, classification (TSC) and forecasting (TSF). The study covers 18 datasets and 21 baselines, providing extensive ablations regarding the pretraining method, imaging alignment method, architectures of the model etc. It concludes that LVMs can transfer well to high-level pattern-centric TSC, but face challenges in low-level, numeric TSF. Fine-tuning analysis show that for TSC, updating MLP and norm layer proves to be the most efficient manner, while for TSC, norm-only tuning helps most.

Strengths:
1. The paper is well-written and easy to follow. The structure is very clear. The highlights and key insights are very helpful.
2. The paper includes comprehensive and careful benchmarking across models, imaging pipelines, datasets and ablations. The experiments give very convincing insights into the future applications of LVM for TS analysis.
3. The identification of inductive biases (periodicity with UVH) and limits with long contexts are very important. It can well motivate concrete future work.

Weaknesses:
1. One significant limitation is that the evaluated LVMs are somewhat outdated. The supervised baseline is trained on ImageNet, which is typically not considered a large foundation model. The self-supervised models are also dated and homogeneous, both MAE and SimMIM rely on masked image modeling. Incorporating experiments on newer models with more diverse training objectives, such as SAM[1], DINOv2, and MLCD [2], would make the results more convincing and impactful.
2. The conclusion of TSF relies heavily on UVH and periodic datasets such as ETT, Electricity, and Traffic. It is not clear whether the conclusion still holds for weak-periodicity datasets.
3. In the main text, the conclusions are drawn based on metrics averaging across horizons and datasets. This can mask per-horizon behaviors. Per-horizon statistical tests would strengthen the claims.
4. The paper does not discuss how to use LVM in TSF with exogenous variables.
5. While I recognize that this is a benchmarking rather than a methodology paper, there are more principled ways to adapt LVMs for time-series modeling. For instance, prior work has adapted ViTs pretrained on images to capture temporal information in videos [3]. If the goal is to leverage LVMs for time-series analysis, a carefully designed adaptation pipeline of this kind may be preferable. How, then, do the authors envision their conclusions transferring to, or motivating, the design of such adaptation methodologies?

[1] Kirillov, Alexander, et al. "Segment anything." Proceedings of the IEEE/CVF international conference on computer vision. 2023.
[2] An, Xiang, et al. "Multi-label cluster discrimination for visual representation learning." European Conference on Computer Vision. Cham: Springer Nature Switzerland, 2024.
[3] Yang, Taojiannan, et al. "Aim: Adapting image models for efficient video action recognition." arXiv preprint arXiv:2302.03024 (2023).

**Audience:**

Yes

**Audience Explanation:**

The paper tackles an important and timely question: whether pretrained LVMs are useful for time-series analysis. Many of its conclusions are intriguing and have the potential to inspire new methodologies and applications.

**Claims And Evidence:**

Yes

**Claims Explanation:**

The paper presents comprehensive experiments across multiple datasets and models and draws several conclusions about the suitability of LVMs for time-series analysis. These conclusions are consistently supported by solid empirical evidence.

**Requested Changes:**

1. In Section 4.2, please clarify the exact training and fine‑tuning protocol, whether the visual encoders were entirely frozen or if specific modules were updated.
2. What results would we see if the vision encoders were adapted with parameter‑efficient methods such as LoRA, adapters, or instruction/prefix tuning?
3. For RQ4, an important missing experiment is to disentangle the effects of fine‑tuning the positional encodings and the patch embedding layer. Because time series and images are different modalities, these components may be the primary adaptation bottlenecks.
4. How is the variate order determined for UVH and MVH, and how sensitive are the results to this choice? Please clarify whether the reported metrics correspond to a particular ordering, the average over multiple random permutations, or the best observed order, and include a brief robustness check if feasible.
5. The inputs are normalized using instance‑level mean and variance; this may suppress magnitude information that can matter for forecasting. For TSF, do you employ any de‑normalization mechanism (e.g., a RevIN‑style layer) to restore the original scale at inference? Since image models often assume global dataset‑level normalization, it would be useful to compare instance‑level, dataset‑level, and per‑variate normalization to determine which is most appropriate here.
6. In the encoder‑only setup (Fig. 1c), to which tokens is the linear head applied: the CLS token, a pooled representation over patch tokens, or the patch tokens aligned with the most recent timestamp? A brief ablation and a clear description of how tokens map back to timesteps would remove ambiguity about the decoding pathway.
7. Please add a brief section comparing LVMs, LLMs, and native time‑series foundation models for time‑series tasks, with guidance on when each is most appropriate.

---

### Review · Reviewer_1EVS · 2026-06-22

**Summary Of Contributions:**

This paper provides a systematic approach to studying the feasibility of LVMs in time series analysis using cross-modal knowledge transfer approaches. In this study, four state-of-the-art LVMs (Vision Transformer (ViT), Swin, Masked Autoencoders (MAE), and SimMIM)) are compared with 21 other models on 18 benchmarks using eight different techniques that convert the time series into images. Two types of problems, namely pattern recognition tasks (Time Series Classification – TSC) and numerical inference tasks (Time Series Forecasting - TSF), are explored. Through extensive ablation experiments, the best combinations of LVMs (e.g., ViT and Gramian Angular Field - GAF for TSC and MAE and UVH for TSF) are obtained.

**Key Strengths**:

- Benchmarking on a Large Scale: The scope of this benchmark is impressively broad as it gives us a comprehensive assessment of different imaging approaches and LVM architectures using standard datasets.

- Component-wise Analysis of Value: Ablation analysis provides very insightful observations, particularly the surprising one that pretrained decoders add far more value to the TSF problem compared to pretrained encoders.

- Highlighting Inductive Bias: This paper has been able to formally define and empirically validate the inductive bias of different imaging approaches, especially the fact that UVH biases our model towards using periodicity while forecasting.

**Key Weaknesses**:

- Constraints on Novelty in Architectures: This is a clear limitation in the paper, where the researchers use already available LVM architectures and imaging techniques without proposing any new architectures.
- Performance on Datasets with Non-Periodic Information: This is another concern raised by the analysis, which highlights the difficulty of the best-performing LVM architecture (MAE+UVH) in performing better than other non-LVM baselines on non-periodic datasets (such as Weather, Illness, and Exchange).
- Scalability Constraints: The scalability of the proposed models to use long look-back windows (more than ~1000 time steps) is limited due to the constraints of the number of pixels in the pre-trained vision architecture and partial utilization of pixels in the imaging step.

**Additional Comments:**

In all, this is a very readable paper. The use of the critical difference plots makes the ranking of the imaging techniques quite easy to understand, while the distinction between high-level (pattern) and low-level (numeric) tasks is a great way to introduce the problem. There is a lot of information presented in the appendices; well done.

**Audience:**

Yes

**Audience Explanation:**

The connection between foundation models and time series analysis is currently a highly active area of research. Scientists involved in cross-modal knowledge transfer, multimodal learning, and time series forecasting will certainly benefit from the findings of the benchmark and the suggestions for using LVMs in practice. The findings that the pre-trained decoder is essential for performing numerical inference and that LVMs are limited when dealing with a long look-back window provide very specific insights into the existing problems and the future directions of research. This is precisely what the TMLR community expects.

**Broader Impact Concerns:**

This paper involves fundamental benchmarking and does not involve any applications. But time series forecasting is extensively applied in areas which have critical implications, such as health care (for example illness data set) and critical infrastructures (for example electricity and traffic data sets). As the authors mention, SOTA LVM based forecasting models have a strong tendency of inductively learning the previously observed periodic components. If these models are used without any precaution in critical infrastructures, it could lead to severe forecasting errors in case of any irregular behavior. A small statement on broader impacts in terms of ethical implications should be added to the paper.

**Claims And Evidence:**

Yes

**Claims Explanation:**

Supporting evidence for the claims is provided in great detail and rigorously. The authors use 10 well-known UEA datasets for the purpose of classification and 8 benchmark datasets for forecasting tasks to validate the proposed techniques. Claims about the usefulness of the pre-trained weights are well-supported via ablations. In particular, the comparison between LVMs and models trained without pre-training (w/o-LVM and single randomly initialized multi-head attention layer - LVM2ATTN) successfully disentangles the effect of pre-training from that of model architecture complexity. The temporal perturbation experiments (such as shuffling and masking) strongly support the claim that the proposed LVMs extract true temporal dependencies instead of exploiting spurious spatial patterns. Finally, the inductive bias mathematically formalized in Lemma 4.1 gives solid theoretical support to the empirical performance drops caused by inappropriate choices of the segment length relative to dataset periodicity.

**Requested Changes:**

**Critical Adjustments**:

None. The experimental design is sound, and the claims are appropriately scoped and supported by the current evidence.

**Adjustments to Strengthen the Work**:

- Expand on Computational Overhead: Though the training and inference times have been given in Table 5, it would help if more focus was laid on the memory requirement for converting long and multivariate time-series data into high-resolution images.

- Main Text Discussion on Weak Periodicity: Restrictions on the use of UVH image technique for data sets without any clear periodicity are discussed in Appendix B.10. A short discussion about this restriction in the body of the text, in particular, Section 4.4 or in the conclusion will provide a better understanding about the predictive power of LVMs.

- Clarify Look-back Window Limits: As indicated by the authors in section 4.4, there exist performance plateaus once the look-back window has 1000 steps. In this case, it will be good if it can be stated explicitly whether this is an absolute limit due to the 224 X 224 resize limit of the images due to ViT/MAE or due to the alignment constant c=0.4

---

### Review · Reviewer_PxKz · 2026-07-20

**Summary Of Contributions:**

The paper provides a broad benchmark of pretrained visual encoders for time-series classification and forecasting, comparing four vision backbones, eight ways of converting signals into images, 18 datasets, and many non-vision baselines. Beyond reporting performance, it studies how pretraining, imaging choices, fine-tuning, architecture, temporal order, and computational cost affect results, and finds that vision models work well for classification but are less reliable for forecasting. Its main forecasting insights are that self-supervised models and value-preserving heatmaps work best, pretrained decoders matter more than encoders, and current approaches are biased toward periodic signals and do not fully benefit from very long history windows.

The studied topic is rare and important, where the paper goes beyond a simple leaderboard comparison and provides a fairly systematic set of ablations on imaging methods, fine-tuning choices, encoder–decoder roles, temporal perturbations, and computational cost. For the ablation studies, where I particularly like the masking and shuffle experiments. This makes the study practically useful for understanding when image-pretrained models help and where the gains come from. So does the limitation, like long history windows.

Besides, I also appreciate all the methods listed in Figure 1 (a) are evaluated and compared somewhere in the paper.

For the weakness, I kindly refer to the below sections, where the scope and some experimental details.

**Audience:**

Yes

**Audience Explanation:**

I think some individuals may be interested in the numerical time-series forecasting part. It is different from many real-world event/probabilistic forecasting.

**Claims And Evidence:**

No

**Claims Explanation:**

The experiments only cover four relatively conventional standalone vision backbones—ViT, Swin, MAE, and SimMIM—not common modern VLMs/LMMs such as Qwen-VL, GPT-4o, Gemini, or Claude. Therefore, the conclusions about the usefulness and limitations of “large vision models” for time-series analysis feel somewhat over-generalized. Those vision language model are also large vision models, I do not feel large vision model will explicitly exclude them.

Besides, some interpretations are stronger than what the ablations directly support. For instance, reinitializing the encoder or decoder introduces multiple confounding factors beyond the component being studied. The claim that performance stops benefiting from look-back windows beyond around 1,000 steps is also not fully consistent across the reported datasets. These results are interesting, but more controlled experiments are needed to make the central claims convincing.

**Requested Changes:**

I think the first one is the title, the current title directly pull me into topics beyond the scope of this paper. Also, in their FIgure 1, it mostly used the vision encoder part.

I like Figure 7 as the inference time is very important, while is there a more finegrained version, with more models and different settings involved?

---

### Comment · Action_Editor_1Td7 · 2026-07-20
**After review AE comment**

Dear authors and reviewers,

thank you all for the thoughtful and detailed reviews. The reviewers generally find the paper interesting and valuable, particularly the systematic benchmarking and the analysis of when vision-pretrained models help or fail for time-series tasks.

I encourage the authors to respond to the reviewers’ comments in the following two weeks and clarify the points raised. In particular:

 - the scope of the term “Large Vision Models” and whether the current experiments support the breadth of the claims in the title and abstract,
 - the extent to which the conclusions about forecasting limitations, periodicity bias, and long look-back windows should be generalized beyond the evaluated settings,
 - clarifications regarding experimental protocols (fine-tuning details, normalization/de-normalization, token usage, imaging choices),
 - the feasibility and importance of additional analyses suggested by reviewers, such as computational considerations and robustness checks.

Reviewers, please consider the authors’ responses and further discuss any points where additional clarification would affect your assessment.